# Improving Regret Approximation for Unsupervised Dynamic Environment Generation

**Harry Mead**
University of Oxford

**Bruno Lacerda**
University of Oxford

**Jakob Foerster**
University of Oxford

**Nick Hawes**
University of Oxford

## Abstract

Unsupervised Environment Design (UED) seeks to automatically generate training curricula for reinforcement learning (RL) agents, with the goal of improving generalisation and zero-shot performance. However, designing effective curricula remains a difficult problem, particularly in settings where small subsets of environment parameterisations result in significant increases in the complexity of the required policy. Current methods struggle with a difficult credit assignment problem and rely on regret approximations that fail to identify challenging levels, both of which are compounded as the size of the environment grows. We propose Dynamic Environment Generation for UED (DEGen) to enable a denser level generator reward signal, reducing the difficulty of credit assignment and allowing for UED to scale to larger environment sizes. We also introduce a new regret approximation, Maximised Negative Advantage (MNA), as a significantly improved metric to optimise for, that better identifies more challenging levels. We show empirically that MNA outperforms current regret approximations and when combined with DEGen, consistently outperforms existing methods, especially as the size of the environment grows. We have made all our code available here: `https://github.com/HarryMJMead/Dynamic-Environment-Generation-for-UED`.

## 1 Introduction

Deep Reinforcement Learning (RL) has been effective in training highly-capable agents in a number of different challenging settings, such as in real-world robotics applications [1, 2, 48, 28], or games such as Go [52], Chess [53], Starcraft [58] and Dota [5]. However, these deep-RL agents tend to exhibit poor generalisation when transferred to tasks or environments with only small changes to those used to train on [62, 9].

In order to address this lack of robustness, domain-randomisation (DR), training over a diversity of environment parameterisations, has proven successful in a number of applications. However, DR relies on random parameterisations resulting in useful training examples, and in complex environments this may not be the case. Automated Curriculum Learning (ACL) [16, 42] methods aim to produce adaptive curricula for training that ensure the generation of useful training examples whilst maintaining a sufficiently diverse distribution over these environment parameterisations. These methods have shown success over naive domain-randomisation approaches [43, 38].

However, manually designing a suitable curriculum for learning may in itself be a challenge, whilst also limiting the capacity for open-ended learning [59, 60]. Recent work has focused on Unsupervised Environment Design (UED) [11], which has emerged as a widely applicable curriculum design method as no prior environment knowledge is required. In the UED literature, each parameterisation of the environment is referred to as a level, and so UED frames the curriculum design problem as the interaction between a teacher agent designing levels and a student agent training on these levels. The majority of existing work focuses on maximising student regret [11, 27, 40, 8], as prior work [11] has shown that if the student and teacher reach a Nash equilibrium of a minimax regret game, the

39th Conference on Neural Information Processing Systems (NeurIPS 2025).

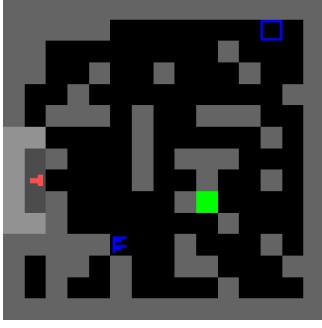 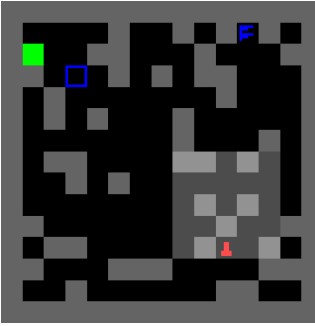

(a) No Key Required     (b) Key Required

Figure 1: Examples of two possible randomly generated levels. In the first, the agent (red triangle) can simply navigate to the goal (green square), whereas in the second, it is required to first obtain the key (two blue triangles) in order to unlock the door (blue unfilled square) blocking the path to the goal

student must necessarily be able to solve all solvable environments. However, computing regret is intractable for many complex tasks, so these methods require *regret approximations*.

Generally UED methods can be categorised as either relying on a learnt level generator [11, 35, 3], or a curation process that selects and replays levels from a randomly generated set [25, 27, 40]. Existing UED methods have focused on environments such as minigrid [11, 8] or bipedal walker [40], where there is a relatively smooth transition in difficulty between levels. However, there are many environments where a small subset of paramaterisations may induce a step-change in difficulty for the student. For example, in the level shown in Figure 1(a), the addition of the door and key to the level generation have no effect on the difficulty of the level for the student agent. The key can simply be ignored and the door acts as any other wall, and so the agent is able to navigate to the goal directly. However, in Figure 1(b), the door blocks the agent's path to the goal, so it is necessary for the agent to first find the key before being able to unlock the door and reach the goal. In this example, these key-requiring mazes represent a very small subset of possible random levels, but they also represent the levels potentially most difficult to learn. Thus, for UED to be effective in environments such as these, it is necessary for methods to identify and train on this more challenging subset of levels.

Whilst replay-based methods are sufficient in small environments, the challenge of sampling and identifying more difficult subsets of levels is amplified as the size of the environment increases. Due to their reliance on random level generation, we show that replay methods fail in larger environments, and thus it is necessary to use learnt level generators. However, training a generator that generates a full level prior to student rollouts presents a challenging credit assignment task, given the long time horizon and sparse rewards. In order to address these challenges with learnt level generation, we propose Dynamic Environment Generation for Unsupervised Environment Design (DEGen). Our method involves dynamically generating the environment as the student agent explores the level, enabling a much denser teacher reward signal, and reducing the difficulty of credit assignment.

However, we show in this work that current regret approximations are insufficient, both for identifying these most difficult subsets of levels and for use in training level generators. We propose Maximised Negative Advantage (MNA) as a more effective regret approximation and show substantial empirical performance improvements over existing regret approximation metrics. Using MNA, we show that DEGen performs substantially better than existing generators that rely on full level generation upfront. We show that DEGen is capable of matching or exceeding the performance of existing replay methods in small environments, unlike previous learnt generators, but performs substantially better as the size of the environment increases. We also show that MNA consistently improves performance over current regret approximations for all UED methods.

Our contributions are:

- We introduce *Dyanamic Environment Generation for UED* (DEGen) as a new method of environment generation, showing performance improvements over existing learnt generators in small environments and significant performance improvements over all methods in larger environments.
- We introduce *Maximised Negative Advantage* (MNA) as a new regret approximation and show substantial improvements over existing metrics.

## 2 Background

### 2.1 Unsupervised Environment Design

Given a specific environment, we can model a level as a Partially Observable Markov Decision Process (POMDP). POMDPs can be defined by a tuple $\langle S, A, O, \mathcal{T}, \mathcal{I}, \mathcal{R}, \rho_0, \gamma \rangle$, where $S$, $A$ and $O$ are the set of states, actions and observations respectively, $\mathcal{T} : S \times A \rightarrow S$ is the transition function, mapping a state-action pair $(s_t, a_t)$ to the subsequent state $s_{t+1}$, $\mathcal{I} : S \rightarrow O$ is the observation function that maps a given state to an observation, $\mathcal{R} : S \times A \rightarrow \mathbb{R}$ is the reward function, $\rho_0$ is the distribution over initial states, and $\gamma$ is the discount factor.

In order to extend this formulation to the framework of UED, the Underspecified POMDP (UPOMDP) is introduced [11], defined by the tuple $\mathcal{M} = \langle S, A, O, \mathcal{T}^\mathcal{M}, \mathcal{I}^\mathcal{M}, \mathcal{R}^\mathcal{M}, \rho_0^\mathcal{M}, \gamma \rangle$. The UPOMDP formulation introduces $\Theta$, the set of all possible free environment parameters $\theta$, for which a specific $\theta$ results in the environment configuration defined by the POMDP $\mathcal{M}_\theta$ with the transition, state and reward functions $\mathcal{T}^\theta, \mathcal{I}^\theta, \mathcal{R}^\theta$ and the initial state distribution $\rho_0^\theta$.

Generally, the UED objective is to identify training levels that maximise the student's regret, given the current student policy $\pi$. The regret is defined as:

$$\text{Regret}(\pi, \theta) = -U(\pi, \theta) + U(\pi_\theta^*, \theta) \tag{1}$$

where $\pi_\theta^*$ is the optimal policy given $\theta$, and $U(\pi, \theta) = \mathbb{E}_{\pi, \mathcal{M}_\theta} \left[ \sum_{t=0}^T \gamma^t r_t \right]$, or the expected discounted return of the policy $\pi$. As such, UED can be framed as a two player minimax regret game:

$$\min_{\pi \in \Pi} \max_{\theta \in \Theta} \text{Regret}(\pi, \theta). \tag{2}$$

By framing UED as this regret-based minimax game, if the environment satisfies the reward conditions outlined in [11], we can guarantee that if the student and teacher policies reach a Nash equilibrium, then the student policy must necessarily be capable of solving all solvable levels.

### 2.2 Existing UED Methods

Whilst the optimal objective shown in Equation 1 has robustness guarantees, in practise, it is infeasible for UED given $\pi_\theta^*$ is required. UED methods such as PAIRED [11] or CLUTR [3] introduce an additional antagonist agent, and regret is approximated as the difference between the performance of the antagonist and student policies. Both these methods rely on RL-trained teacher, where the teacher aims to maximise the performance difference between the antagonist and the student. However, these RL-trained teachers tend to struggle with maintaining diversity over training environments [25]. Some techniques have shown to improve performance, such as behaviour cloning between the antagonist and protagonist and the use of high entropy coefficients [35]. However these RL-based methods still tend to be outperformed by replay-based methods.

Rather than relying on a learnt generator, PLR [27], relies on maintaining a replay buffer of high-regret levels that have been sampled from a random generator. PLR alternates between sampling new random levels and replaying previously sampled levels. PLR relies on a score function to approximate regret, with the two most commonly used score functions being *Positive Value Loss* (PVL) and *Maximum Monte Carlo* (MaxMC) [25, 40].

PVL is defined as

$$\frac{1}{T} \sum_{t=0}^T \left( \max(0, \sum_{k=t}^T (\lambda\gamma)^{k-t} \delta_k) \right) \tag{3}$$

---

**Algorithm 1** DEGen

---

**Initialise:** student policy $\pi_{\phi_1}$, generator policy $\Lambda_{\phi_2}$
**while** not converged **do**
   // Sample $N$ trajectories
   **for** $n \in 1 : N$ **do**
      Initialise empty level
      // Take $T$ student steps
      **for** $t_s \in 1 : T$ **do**
         // Generate partial level
         Sample $\Lambda$ actions to generate section of level that has been observed but not generated
         // Take student action
         Sample $\pi$ action
      **end for**
      compute score using student trajectory $\tau_s$
      assign reward to generator trajectory $\tau_g$
   **end for**
   Update $\phi_1$ according to sampled student trajectories
   Update $\phi_2$ according to sampled generator trajectories
**end while**

---

where $\gamma$ is the discount factor, $\lambda$ is from the Generalised Advantage Estimator [50] and $\delta_t$ is the 1-step TD-error at timestep $t$. We can view PVL as approximating regret as the average advantage, but with the advantage clipped at 0. As such, maximising PVL can be seen as effectively maximising states where the student does better than expected, which intuitively appears to be mismatched with the regret objective. Despite this, empirically, PVL has been shown to be effective.

MaxMC approximates regret using the maximum achieved return ($R_{max}$) on a given level, and is defined as

$$\frac{1}{T} \sum_{t=0}^{T} \left( R_{max} - \hat{V}(s_t) \right) \tag{4}$$

where $\hat{V}(s_t)$ is the learnt value function approximation for the value of the current policy at state $s_t$. MaxMC appears a more intuitive approximation for regret than PVL. However by relying on a Monte Carlo approximation for regret, MaxMC requires a sufficient number of trials in a level such that $R_{max}$ is a good approximation for the optimal return. Additionally, MaxMC can only be used in environments where reward is obtained at the final step, as $R_{max}$ is dependent on the full episode reward.

Whilst PLR has been shown to be effective in a number of domains, relying on random level generation necessarily means that no insight is gained from past levels to influence future levels generation. ACCEL [40] addresses this by augmenting PLR such that new levels are generated by mutating existing levels previously in the replay buffer. This evolutionary approach enables some capacity for learning from previously identified high-regret levels, but still relies on random mutations for new level generation.

An alternate approach for UED is Sampling for Learnability (SFL) [49]. Similarly to PLR, SFL relies on sampling a set of randomly generated levels and selecting those with the highest scores for training. However, rather than this score metric approximating regret, SFL aims to train on levels with high learnability, defined as $p(1 - p)$, where $p$ is the success rate of the current policy on the sampled level.

## 3 Dynamic Environment Generation

Existing work [25, 40, 35] has shown that replay-based UED methods generally outperform methods relying on a learnt generator. This can be attributed to a number of factors, but primarily, level generation presents difficulties for learning. The level generation learning problem has a long time horizon and sparse rewards, and so credit assignment is challenging. Current UED have focused on relatively small environments where it is feasible to sample useful training levels from random level generation. However, as the size of the environment grows, and especially with environments

with features that can add complexity, it becomes more difficult to sample useful levels. Therefore, it becomes necessary to use a learnt generator instead.

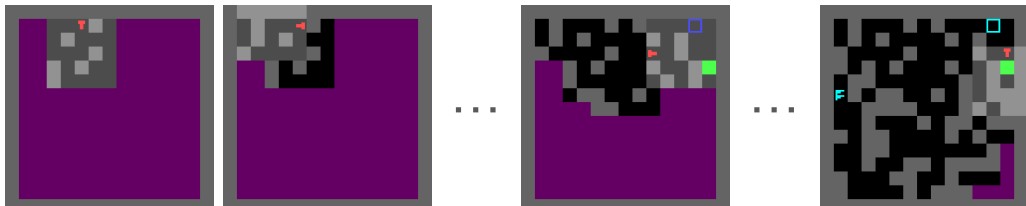

Figure 2: Illustration of how DEGen generates the level as the student agent explores the level, with purple colouring indicating areas that are yet to be observed.

In order to address this, whilst also reducing the challenges inherent to RL-based learnt generators, we propose Dynamic Environment Generation (DEGen), outlined in Algorithm 1. Rather than simply generating the entire environment level initially, we exploit the partial observability of the student agent and instead only generate those parts of the level the student observes, as illustrated in Figure 2. This allows for a much denser reward signal, reducing the difficulty of credit assignment.

If we consider the PVL and MaxMC regret approximations shown in Equations 3 and 4 respectively, both can be written in the form

$$\text{Regret} \approx \frac{1}{T} \sum_{t=0}^{T} G_t, \tag{5}$$

where we approximate regret as the mean of some value $G_t$ across the student trajectory. If the entire level is generated initially, this approximation must simply be assigned as the reward for the last step in the level generation trajectory, resulting in a very sparse reward signal for the generator. However, if the level is generated as the student trajectory unrolls, we are able to assign a much denser generator reward. If we have the function $t_s = \mathcal{T}(t_g)$ that maps the generator timestep $t_g$ to the student timestep $t$, we can instead assign the generator reward $(r_g)$ at timestep $t_g$ as

$$r_{t_g} = \frac{1}{T} \sum_{t=\mathcal{T}(t_g)}^{\mathcal{T}(t_g+1)-1} G_t. \tag{6}$$

This increased reward density reduces some of the difficulty of the credit assignment challenge for training the level generator. Additionally, DEGen reduces noise in the credit assignment by only generating parts of the level the student observers. If the entire level is generated initially, it is likely that some parts of the level will never be observed by the student, and so have no effect on the score of the level. As such, these actions only serve to add additional noise to the credit assignment task, an issue which is negated by DEGen.

Another issue present in existing RL-based generators is a lack of diversity in generated levels. Previous work [35] has shown that a high entropy coefficient can reduce this issue, however we still observed reduced level diversity with existing RL-based generators. As the student policy $\pi$ is stochastic, by generating the level based on where the student explores, we introduce an greater degree of stochasticity in the level generation, which increases the diversity of generated levels. We also found that introducing some additional randomness in level generation - specifically randomly initialising the starting location of the student agent - improved training level diversity, and zero shot agent performance.

## 4 Maximised Negative Advantage

Whilst the PVL and MaxMC regret approximations have shown success in a number of domain, there are flaws with both metrics in relation to generating challenging levels. For PVL, high positive advantage will result in a high PVL score, whereas generally, more challenging levels would tend to be more difficult than expected, and so more likely to result in high negative advantage. For MaxMC,

high scores require at least one high return rollout, and again, for the most challenging levels, this is unlikely to occur. Whilst PVL and MaxMC have shown success with use in replay-based methods, we observed poor performance when these metrics were directly optimised for, shown in Appendix D.2. We instead propose Maximised Negative Advantage (MNA) as a more suitable metric.

Consider the policy $\pi$ with the value-function $V(s)$. We are aiming for an approximation of Equation 1, requiring an approximation for both $U(\pi_\theta^*, \theta)$ and $U(\pi, \theta)$. If we assume a deterministic environment, given a trajectory of length $T$, if $V(s)$ is the true value function, we can lower bound $U(\pi_\theta^*, \theta)$

$$U(\pi_\theta^*, \theta) \geq \max \begin{pmatrix} V(s_0), \\ \gamma V(s_1) + r(s_0, a_0), \\ \vdots \\ \gamma^T V(s_T) + \sum_{k=0}^{T-1} \gamma^k r(s_k, a_k) \end{pmatrix}. \tag{7}$$

We label this maximum over value functions

$$V_n^{\max}(s_t) = \max \begin{pmatrix} V(s_t), \\ \vdots \\ \gamma^n V(s_{t+n}) + \sum_{k=t}^{t+n-1} \gamma^{k-t} r(s_k, a_k) \end{pmatrix}. \tag{8}$$

Given this, as the true value function $V(s_0)$ gives us the expected performance of the current policy, and the maximum over value functions $V_T^{\max}(s_0)$ lower bounds the performance of the optimal policy, we can lower bound the regret as

$$\text{Regret} \geq -V(s_0) + V_T^{\max}(s_0). \tag{9}$$

However, in practise, the exact value function $V(s)$ will generally be unknown, and instead must be approximated with a learnt value function $\hat{V}(s)$. As this learnt value function may overestimate the value of the state, the inequality in Equation 7 does not necessarily hold when $V(s)$ is replaced with $\hat{V}(s)$. Additionally, the $\hat{V}(s_0)$ approximation for $U(\pi, \theta)$ will be biased, being a learnt value approximation. We can reduce the likelihood of $\hat{V}_T^{\max}(s_0)$ exceeding $U(\pi_\theta^*, \theta)$ by instead using the approximation $\hat{V}_n^{\max}(s_0)$, where $n < T$, although this instead results in a potentially overly conservative approximation. Similarly, we can reduce the bias of our approximation for $U(\pi, \theta)$ by instead using the approximation $\gamma^n V(s_n) + \sum_{k=t}^{n-1} \gamma^k r(s_k, a_k)$, however this then introduces greater variance. We therefore define the n-step regret approximation at timestep $t$ as

$$\hat{G}_t^{(n)} = -\left(\gamma^n V(s_{t+n}) + \sum_{k=t}^{t+n-1} \gamma^{k-t} r(s_k, a_k)\right) + \hat{V}_n^{\max}(s_t) \tag{10}$$

and we note the similarity between this regret approximation and the negative n-step advantage estimation [50]. In order to balance both the bias and variance of the $U(\pi, \theta)$ approximation, and the conservativeness of the $U(\pi_\theta^*, \theta)$ approximation, in line with the Generalised Advantage Estimator [50], we introduce the regret approximation

$$\hat{G}_t^\lambda = (1 - \lambda) \sum_{n=0}^\infty \lambda^n \hat{G}_t^{(n)}. \tag{11}$$

Empirically, we find that rather than just approximating regret at the first state, using the mean regret approximation was a more effective metric

$$\frac{1}{T} \sum_{t=0}^T \hat{G}_t^\lambda. \tag{12}$$

We show empirically that this regret approximation is much suitable optimisation metric for learnt generators, whilst also showing improved performance when used in replay-based methods.

## 4.1 Solvability

Whilst the metric in Equation 12 does allow for more challenging levels to be sampled than existing metrics, the reliance on the learnt value function $\hat{V}(s)$ to determine maximum possible performance does present issues with ensuring level solvability. If the environment satisfies the reward conditions [11] that ensure the teacher-student regret Nash equilibrium results in an student policy capable of solving all possible solvable levels, then regret is maximised by generating solvable levels. Therefore, a good regret approximation metric should not result in high scores for unsolvable levels. Both PVL and MaxMC implicitly score low for unsolvable levels. For the Nash equilibrium result to hold, the reward conditions [11] necessitate that the maximum achievable return for an unsolvable level $F_{max}$ must not exceed the minimum return achievable in a solved level $S_{min}$. In the case of MaxMC, a higher $R_{max}$ can be achieved if the level is solvable, and so solvable levels will generally score higher. For PVL, higher returns will tend to result in higher advantage, and therefore a higher score, so solvable levels that can achieve higher returns will score high.

The issue with this implicit bias towards solvable levels is that in practice, this manifests as a bias towards levels with a high success rate [49], i.e. the current policy solves the level with a high probability. Therefore, these metrics generally do not produce sufficiently challenging levels for training. SFL [49] has shown that training using levels with approximately $50\%$ success rate results in strong training performance. However, determining the exact success rate for a given level requires substantially more environment rollouts than necessary for metrics such as PVL or MaxMC.

While MNA is capable of identifying challenging levels with significantly fewer rollouts than directly scoring based on success rate, over-approximations of state value $\hat{V}(s)$ may result in high scores for unsolvable levels. In order to compensate for this, we introduce an explicit penalisation for unsolvable levels. As it is often intractable to determine exact solvability of levels in complex environments, we instead introduce approximate unsolvablilty, where we define a level as approximately unsolvable if it has never been solved, e.g. has a success rate of $0\%$. Therefore, if a level is approximately unsolvable, we set the score for the level to zero. As such, our final proposed regret approximation is

$$\text{MNA} = \left( \frac{1}{T} \sum_{t=0}^{T} \hat{G}_t^\lambda \right) \cdot \hat{C} \tag{13}$$

where $\hat{C}$ is 0 if the level is approximately unsolvable and 1 otherwise.

## 5  Experimental Setup

For this work, we examine the standard minigrid environment used in previous UED work [11, 25, 40], as well as evaluating UED performance on the modified minigrid with the addition of a key and locked door. In line with exisiting UED work, our main evaluation metric is zero-shot performance on a set of hand-designed test levels. For the standard minigrid, we use the set of 8 test levels used in previous work [10, 49]. For the modified key minigrid, we modify this set of levels so as to require the agent to unlock the door to reach the goal. Previous work has only examined minigrid when students are trained using 13x13 levels, but in order to scale UED to larger environments, we need to ensure that the student is still capable of learning complex skills, such as unlocking a door, even when trained in larger environments. To assess the student's ability to solve levels that require the door to be unlocked, we evaluate student performance on the existing key minigrid test levels but when trained on levels that are 17x17 and 21x21.

**Baselines:** In order to assess the effectiveness of both MNA and DEGen, we compare a number of different existing level generation methods and a number of regret approximation metrics. For regret approximation metrics, we compare MNA to the existing metrics, MaxMC and PVL. For assessing these metrics, we use the existing UED methods PLR [25] and ACCEL [40]. We include Domain Randomisation (DR) to show the relative performance of UED compared to a naive, random approach. Additionally, we include SFL [49] for a non-regret based approach. For the standard minigrid environment, we also include the Initial Gen baseline, corresponding to an RL-trained teacher that generates the full environment prior to student rollouts, however we show that this performs substantially worse than all other methods so do not include it in the key minigrid domain. All student agents are trained using PPO [51], as well as the teacher agents used in DEGen and Initial

Gen. Detailed training hyperparameters for all domains and UED methods are found in Appendix B.2.

# 6  Results

## 6.1  Minigrid

Figure 3 illustrates the performance of MNA and DEGen compared to existing baselines. From these plots, it is clear that Intial Gen performs substantially worse than all other methods, whereas DEGen performs comparably to existing replay-based methods. This substantial performance deficit can likely be attributed to a lack of diversity in the generated levels, see Appendix D.5. We see a far greater diversity in the levels generated via DEGen, and this, along with the reduced credit assignment challenge, allows for DEGen to substantially outperform Initial Gen, despite the former also relying on an RL-trained teacher. We also see that for both PLR and ACCEL, MNA outperforms the existing MaxMC and PVL regret approximations. However, Figure 3 also shows that in this setting, early in training, DEGen is outperformed by replay-based methods using MNA. This suggests that, whilst the final DEGen performance is equal or greater than the performance of replay-based methods, the additional challenge of learning the generator does impact initial performance.

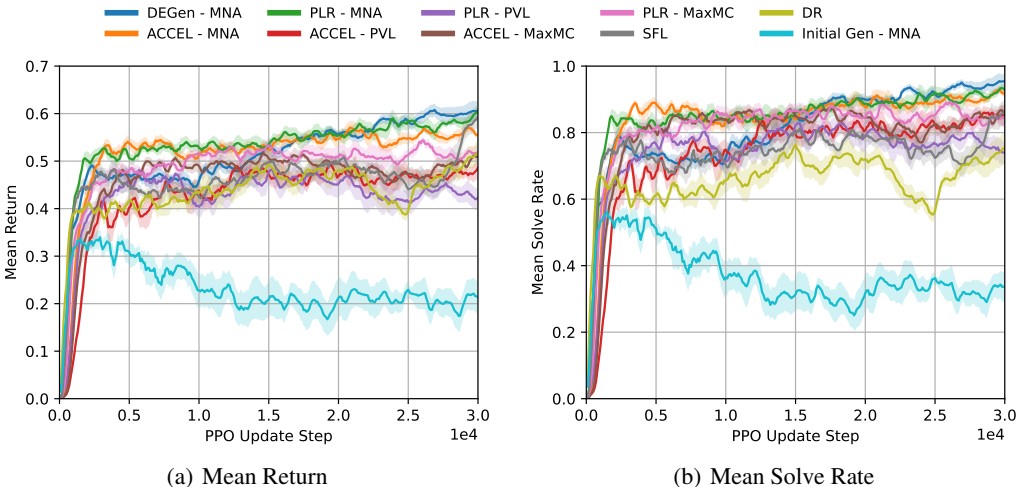

(a) Mean Return                    (b) Mean Solve Rate

Figure 3: Minigrid zero-shot performance on hand-designed test set, showing mean and standard error across 8 runs.

## 6.2  Key Minigrid

Figure 4 compares the performance of the various tested UED methods in the key minigrid domain. Due to the increased challenge of level generation with the addition of a key and locked door, there is far more variance in the relative performance of each of the methods. In this domain, we see DEGen outperform all existing baselines, as well as the MNA-based replay methods. We see that PLR using either MaxMC or PVL performs extremely poorly, whereas PLR using MNA, which is much more capable of identifying challenging levels, is the best performing baseline. This highlights the improved regret approximation of MNA compared to existing metrics. We also note that SFL performs poorly in this key-minigrid setting. Learnability only considers the final outcome of an episode, rather than the full student trajectory, and this result suggests that this may be insufficient in domains where additional environment features result in a more complex subset of levels.

## 6.3  Increased Environment Size for Key Minigrid

Up to this point, we have examined only the relatively small 13x13 environment setting. However, as outlined above, with the aim of scaling UED to larger, more complex environments, we examine the

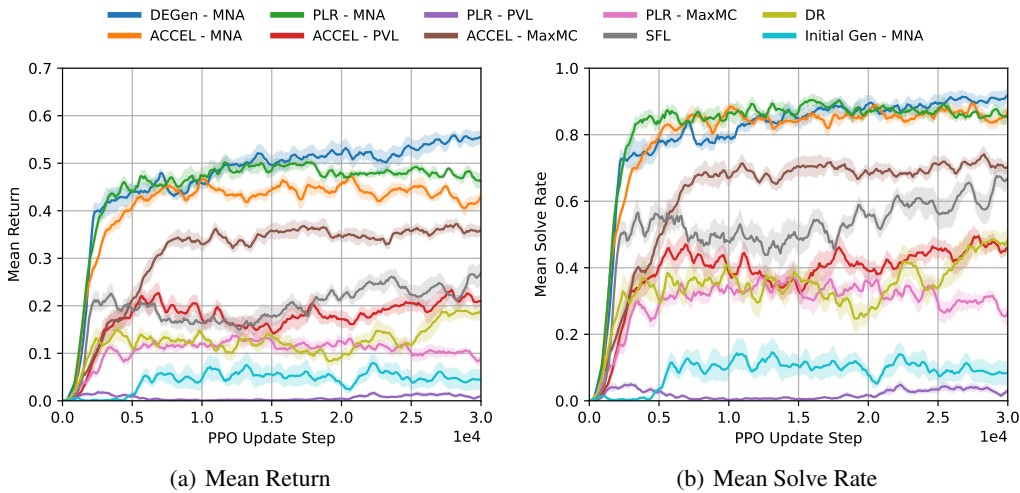

(a) Mean Return          (b) Mean Solve Rate

Figure 4: Minigrid with key and locked door zero-shot performance on hand-designed test set, trained on 13x13 training levels, showing mean and standard error across 8 runs.

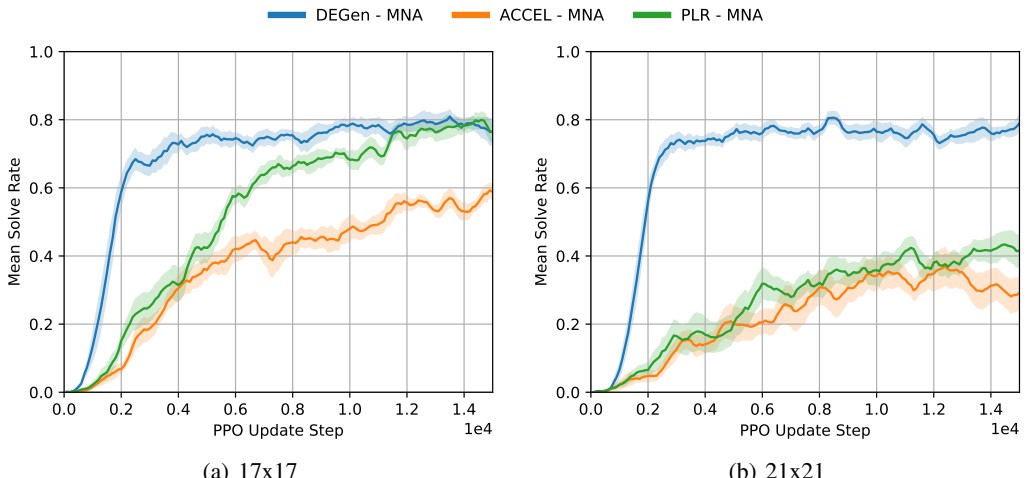

(a) 17x17          (b) 21x21

Figure 5: Minigrid with key and locked door zero-shot performance on hand-designed test set, trained on larger training levels, showing mean and standard error across 8 runs.

performance of current UED methods when simply scaling up a version of minigrid to sizes larger than 13x13. Figure 5 shows the performance of DEGen, PLR and ACCEL using MNA when training levels are 17x17 and 21x21. Whilst there is a performance drop compared to the 13x13 levels, as the size of the environment grows, DEGen substantially outperforms existing methods. This result highlights that, with any domains with additional environment features such as the key and door that can potentially add additional complexity to policy learning, as the environment size grows, it becomes substantially harder to sample useful levels where these features are necessary. Therefore it is necessary to use a trained generator and DEGen is able to overcome the credit assignment challenges present in previous methods for training learnt generators.

## 7 Discussion

It is clear that from these results that DEGen does substantially outperform existing baselines in both the minigrid, and key minigrid environments, and this performance improvement is increased as the

environment size increases. Scaling environment size is likely a necessity if future work aims to move UED from the current set of toy small-scale environments used, to more useful real-world domains, and this work shows that replay-based methods perform poorly with relatively small increases to environment size. We have demonstrated that DEGen is an effective method of addressing these issues, both compared to generating the full level prior to student rollouts, and replay-based methods.

## 8    Limitations

Whilst the results we have presented in this paper do show strong performance from DEGen compared to existing baselines, the domains presented in this paper present relatively simple mappings between the level representation and the agent's current observation. In more complex domains, such as 3D Games [22, 15, 20], or real-world robotics applications [39, 63, 21], it will be more complex to determine how specific environment parameters affect what the agent is currently observing. In order for UED to bridge the gap from the current set of game domains to real-world applications, it would be necessary for DEGen or DEGen-like methods to address this limitation. We believe that World Models [19] represent a promising direction for future research to address this. In its current form, DEGen relies on both the agent and the generator interacting with a fixed environment, where the environment has an explicit mapping between the agent's observation and the level parameters, and the generator is only able to generate the level where the agent has observed. However, a world model guided by a regret approximation such as MNA would represent a generator that could directly generate observations for the agent. Rather than relying on explicit mapping between level and observation, this mapping could be learnt with environment data when training the world model. Whilst the training of a world model would add additional computational cost to the training process, it would enable DEGen methodology to be applied to substantially more complex environments. With the advent of highly general world models such as the Genie series of world models [6], this could represent a path to training highly general policies that are effective in a wide variety of applications.

Additionally, in line with previous work [25, 40, 8], we have compared the relative performance of UED methods based on the number of student PPO update steps. However, training the DEGen teacher agent adds a high computational cost to the training loop, and so training using DEGen takes approximately four times as long as training using methods such as PLR and ACCEL. Full details on compute time and experiment specification can be found in Appendix B. Therefore, replay-based methods may be preferable for small environment sizes where similar performance is achieved. However again, it is clear than as environment size grows, the maximum performance achieved by DEGen exceeds replay-based methods, and so the additional time cost is justified, given the performance gains.

## 9    Conclusion

In this paper, we introduce a new level generation method, Dynamic Environment Generation for UED, and a new regret approximation metric, Maximised Negative Advantage. We outline how current UED methods fail as training environment size increases, and show that DEGen is capable of mitigating the issues associated both with these larger environments, and with RL-based level generation. We show that the use of MNA enables DEGen to outperform existing baselines, whilst also showing that the use of MNA consistently improves the performance of existing UED methods. These performance improvements are most evident in the more complex key minigird domain. We believe there is significant potential for future UED research to address larger and more complex environments, and that approaches based on MNA and DEGen provide a promising foundation for this advancement.

## Acknowledgements

This work was supported by the EPSRC Centre for Doctoral Training in Autonomous Intelligent Machines and Systems [EP/S024050/1]. Lacerda and Hawes have received EPSRC funding via the "From Sensing to Collaboration" programme grant [EP/V000748/1].

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

# A Related Work

This research presents a new method for Unsupervised Environment Design (UED) [11], an area of research focused on improving the generalisation of policies trained using reinforcement learning and enabling open-ended learning. Domain Randomisation (DR) can be categorised as the most basic form of UED, where environment parameters are simply randomised naively for training. DR has shown to be effective in applications such transferring policies from simulations to real world robotics deployments [55, 41, 2]. Prioritised Level Replay (PLR) [27, 25] augments domain randomisation by maintaining a replay buffer of levels that have been scored as effective for training. PLR has been employed in a diverse set of domains, such as in training World Models [45] or meta-reinforcement learning [24]. Additional work has examined dealing with distribution shift between training and deployment [26]. Whilst PLR relies on replaying levels from a fixed random generator, other work has focused on generating new levels through evolving prior levels [40]. POET [59, 60] examines co-evolving both the environment levels, but also a population of agents playing those levels. More similarly to DEGen, other existing work focuses on explicitly learning a level generator [11, 3, 35, 8].

PAIRED [11] introduced the formalisation for UED, framing the level design problem as a minimax regret game between the student and teacher. PAIRED relies on the performance difference between a protagonist and antagonist agent to score level suitability, and uses these scores to train a level generator. CLUTR [3] improves on PAIRED by learning a latent representation of levels to reduce the challenge of learning the generator. ADD [8] has used a diffusion model to generate levels instead. More recent work has examined using non-regret based scoring of levels [57, 56] such as learnability [49, 37, 17].

UED can be seen as a form of Automatic Curriculum Learning (ACL) [43, 16], where ACL aims to provide an automatic curriculum to enable learning of increasingly challenging tasks. Unlike UED, ACL often relies on specific knowledge of the target task [14, 34].

This work also relates to the field of Procedural Content Generation (PCG) [32, 46]. PCG has focused on level design for games, such as with terrain generation [54], task design [12, 31] or puzzle setting [30, 4]. Much of this work has focused on level design for human play, and some of this work relies on specific user input for level generation [33, 7, 44]. Methods using RL for training level generators [29, 13] require hand-designed, environment specific generation rewards, which differs from the domain-agnostic level scoring used in UED.

Unlike previous UED work, DEGen relies on a teacher that generates the environment based on the student's current observations. This shares similarities with existing work in World Models [19]. Whilst the DEGen teacher is able to control the observations of the student, the environment is designed to explicitly disallow infeasible observations to be generated. World Models are instead trained on example trajectories, and the feasibility of generated observations is implicitly learnt. These learnt models are capable of training agents without access to the real environment [47, 36, 20]. Large open-ended world models [61, 23, 6] may potentially enable the training of highly generally capable agents . Recent work has examined applying UED methods, specifically PLR, to training agents in world models [18].

# B   Detailed Experimental Setup

## B.1   Environment Details

### B.1.1   Minigrid

We use the standard minigrid implementation from exisiting UED work [11, 25, 40, 49, 8]. Examples of levels are shown in Figure 9.

**Observations:** The agent, depicted in red, observes a 5x5 square in front of it. It also has access to its absolute direction, e.g. whether it is facing *North*, *South*, *East* or *West*.

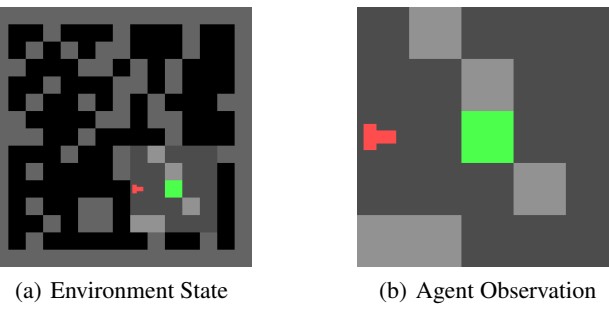

(a) Environment State            (b) Agent Observation

Figure 6: Minigrid Agent Observations

**Rewards:** The agent receives reward $r_g$ if it reaches the goal. The size of this reward is determined by the number of steps taken to reach the goal $T$, and the parameter $T_{\max}$, specifying the maximum number of steps before an episode terminates.

$$r_g = 1 - 0.9 \left( \frac{T}{T_{\max}} \right) \tag{14}$$

**Actions:** At each step, the agent can either move *forward*, turn *left* or turn *right*.

### B.1.2   Key Minigrid

The key minigrid implementation is identical to the standard minigrid level, except for the the addition of a key and locked door.

**Observations:** The agent is additionally able to observe whether it has collected the key.

**Rewards:** The reward remains identical to the standard minigrid, with reward only being received on reaching the goal. Note that there are no specific rewards associated with collecting the key or unlocking the door.

**Actions:** The agent has an additional *use* action. The agent will pick up the key if it reaches the grid square the key is on. In order to unlock the door, the agent must select the *use* action when it has the key and the door is directly ahead of the agent.

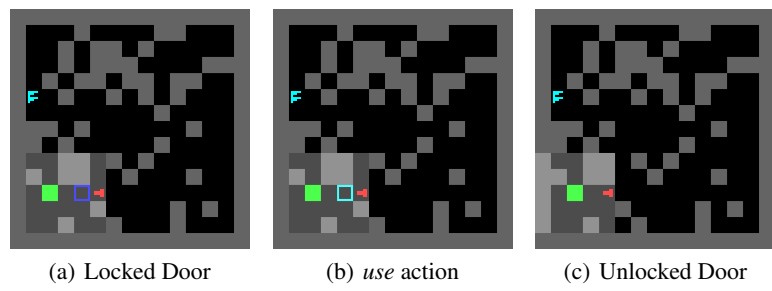

(a) Locked Door            (b) *use* action            (c) Unlocked Door

Figure 7: Key Minigrid Door Unlocking

### B.1.3 DEGen Teacher

**Observations:** The teacher's observations are an augmented version of the student agent. Similarly to the student, the teacher observes a 5x5 square in front of the student. However, this is augmented by an overlayed 5x5 square that indicates which squares have yet to be generated.

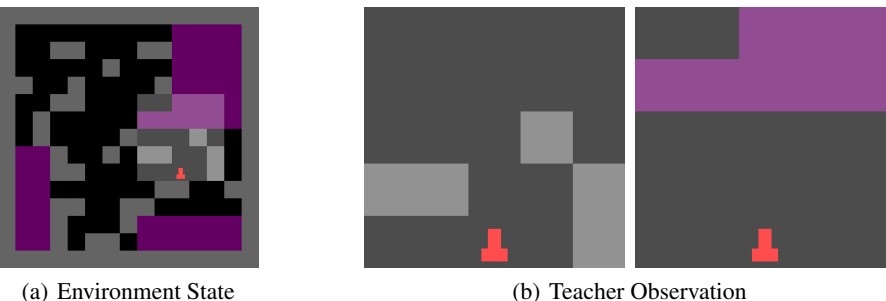

(a) Environment State          (b) Teacher Observation

Figure 8: DEGen Teacher Observations

**Actions:** At each step, the teacher is able to fill any ungenerated grid square the student is currently observing. The teacher action $a$ is composed of two sub-actions $(a_1, a_2)$ where $a_1$ selects which cell will be filled, and $a_2$ selects what the cell will be filled with, e.g. *wall*, *empty*, *goal*, *key*, *door*. We use action masking to ensure that only previously ungenerated cells are filled, as well as enforce that only one *goal*, *key* and *door* can be placed for each level.

**Rewards:** The teacher rewards are determined by the chosen regret approximation. These dense rewards are calculated using Equation 6.

**Dynamics:** As the environment is rolled out, at each step, either the teacher or the student will act. If there are ungenerated cells in the agent's observation, the teacher must fill all ungenerated cells currently observed before the student is able to take another action.

As both the student and teacher have partial observability, both policy networks include an LSTM layer, so actions are conditioned on all previous observations. The teacher is conditioned on all previous observations, whereas the student is conditioned only on the fully-generated observations.

### KL Regularisation

Prior work [35] has shown that entropy regularisation is necessary when using reinforcement learning to train a teacher for UED. The policy gradient loss is augmented with an entropy loss

$$L_{\text{entropy}} = \mathbb{E}_{s \sim \pi}\left[-H\Big(\pi(\cdot|s)\Big)\right]. \tag{15}$$

Entropy regularisation biases the policy towards a more uniform distribution of action probabilities. However, as there can only be a maximum of a single *goal*, *key* and *door* per level, the distribution of cell objects will be highly non-uniform, as the vast majority of cells will be either *wall* or *empty*. Therefore rather than including an entropy loss for the $a_2$ sub-action, we instead add KL regularisation with a fixed categorical distribution $q(a_2)$, defined with the parameter $p_g$, such that

$$q(a_2) = \text{Cat}(\boldsymbol{\alpha}), \qquad \boldsymbol{\alpha} = (p_w,\ p_w,\ p_g,\ p_g,\ p_g), \qquad p_w = \frac{1 - 3p_g}{2} \tag{16}$$

and

$$L_{\text{emtopy} + KL} = \mathbb{E}_{s \sim \pi}\left[-H\Big(\pi_{a_1}(\cdot|s)\Big) + D_{KL}\Big(\pi_{a_2}(\cdot|s)\Big|\Big|q(\cdot)\Big)\right]. \tag{17}$$

For all experiments, we used $p_g = 0.01$.

## B.2 Hyperparameters

Full code and instructions on how to run can be found at `https://github.com/HarryMJMead/Dynamic-Environment-Generation-for-UED`.
All existing methods were trained using implementations based on JaxUED [10], available at `https://github.com/DramaCow/jaxued`, and SFL [49], available at `https://github.com/amacrutherford/sampling-for-learnability`. Learning hyperparameters are shown in Table 1 and the replay UED hyperparameters are shown in Table 2.

Table 1: Learning Hyperparameters.

| Parameter | Minigrid | Key Minigrid 13x13 | 17x17 and 21x21 |
|---|---|---|---|
| **Student PPO** | | | |
| Number of Updates | 30000 | | 15000 |
| $\gamma$ | 0.995 | | |
| $\lambda_{\text{GAE}}$ | 0.95 | | |
| PPO number of steps | 512 | | |
| PPO epochs | 4 | | |
| PPO minibatches per epoch | 4 | | |
| PPO clip range | 0.04 | | |
| PPO # parallel environments | 256 | | |
| Adam learning rate | 5e-4 | | 2.4e-4 |
| Anneal LR | yes | | no |
| Adam $\epsilon$ | 1e-5 | | |
| PPO max gradient norm | 0.5 | | |
| PPO value clipping | yes | | |
| value loss coefficient | 0.5 | | |
| entropy coefficient | 1e-3 | | |
| Hidden dimension size | 256 | | |
| **Teacher PPO** | | | |
| $\gamma$ | 0.998 | | 0.99 |
| $\lambda_{\text{GAE}}$ | 0.95 | | |
| PPO epochs | 4 | | |
| PPO minibatches per epoch | 4 | | |
| PPO clip range | 0.2 | | |
| Adam learning rate | 1e-3 | | |
| Anneal LR | yes | | no |
| Adam $\epsilon$ | 1e-5 | | |
| PPO max gradient norm | 0.5 | | |
| PPO value clipping | yes | | |
| value loss coefficient | 0.5 | | |
| entropy coefficient | 5e-2 | | |
| Hidden dimension size | 256 | | |
| Num Teacher Steps *(Initial Gen)* | 60 | | |

Table 2: UED Hyperparameters.

| Parameter | Minigrid | Key Minigrid |
|---|---|---|
| **PLR** | | |
| Replay rate, $p$ | 0.5 | |
| Buffer size, $K$ | 8000 | |
| Prioritisation | Rank | |
| Temperature, $\beta$ | 1.0 | |
| staleness coefficient | 0.3 | |
| **ACCEL** | | |
| Number of Edits | 20 | |
| Buffer size, $K$ | 8000 | |
| Prioritisation | Rank | |
| Temperature, $\beta$ | 1.0 | |
| **SFL** | | |
| Batch Size $N$ | 25000 | |
| Rollout Length $L$ | 20000 | |
| Update Period $T$ | 100 | |
| Buffer Size $K$ | 1000 | |
| Sample Ratio $\rho$ | 0.5 | |

### B.3 Compute Details

For all experiments, each run was on 1 Nvidia A40. We show the mean compute time for both domains and each UED method in Table 3.

Table 3: Compute Time.

| Method | Compute Time (hh:mm) | |
|---|---|---|
| | **Minigrid** | **Key Minigrid** |
| DR | 11:16 ± 00:02 | 12:01 ± 00:02 |
| Initial Gen | 13:39 ± 00:01 | 13:49 ± 00:02 |
| PAIRED | 23:32 ± 00:02 | 23:33 ± 00:01 |
| SFL | 10:53 ± 00:01 | 10:48 ± 00:02 |
| PLR | 06:53 ± 00:02 | 06:53 ± 00:01 |
| ACCEL | 05:43 ± 00:02 | 05:45 ± 00:01 |
| DEGen | 25:49 ± 00:00 | 25:53 ± 00:01 |

## B.4 Zero-shot Transfer Levels

Figures 9 and 10 show the hand designed levels used for evaluating zero-shot performance. The minigrid levels were taken directly from JaxUED [10]. The key minigrid levels have been modified so that the student is required to unlock the door to reach the goal. Note that we have chosen to include the FourRooms_Key levels rather than modified versions of the labyrinth levels, as the key would trivially be on the path to the goal for these labyrinth levels.

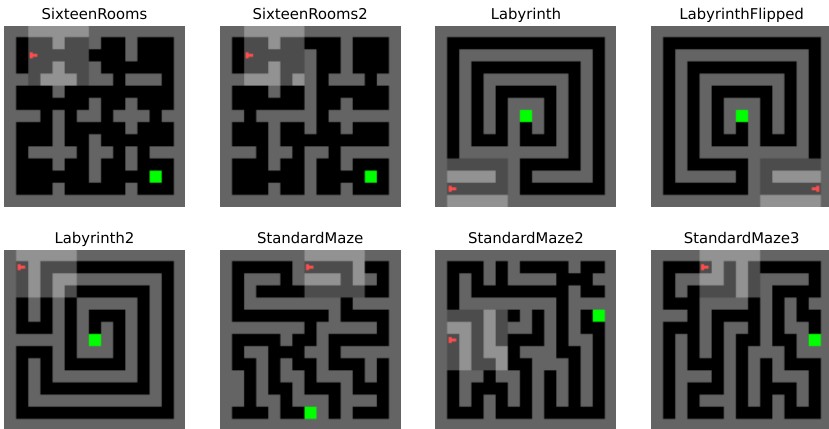

Figure 9: Hand designed evaluation levels for minigird

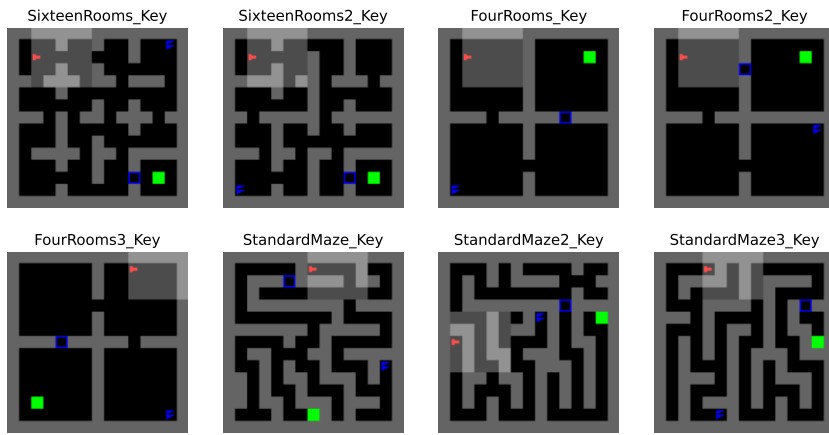

Figure 10: Hand designed evaluation levels for key minigird

## B.5 Tabular Results

**Minigrid Results**

Table 4: Minigrid Solve Rate (1)

| Level | Initial Gen - MNA | DR | SFL | PLR - MaxMC | ACCEL - MaxMC |
|---|---|---|---|---|---|
| SixteenRooms | 0.93 ± 0.06 | **1.00 ± 0.00** | **1.00 ± 0.00** | **1.00 ± 0.00** | **1.00 ± 0.00** |
| SixteenRooms2 | 0.48 ± 0.14 | **1.00 ± 0.00** | **1.00 ± 0.00** | **1.00 ± 0.00** | **1.00 ± 0.00** |
| Labyrinth | 0.04 ± 0.04 | 0.51 ± 0.16 | 0.60 ± 0.13 | 0.45 ± 0.12 | 0.69 ± 0.13 |
| LabyrinthFlipped | 0.11 ± 0.10 | 0.38 ± 0.15 | 0.68 ± 0.13 | 0.64 ± 0.12 | 0.50 ± 0.13 |
| Labyrinth2 | 0.05 ± 0.04 | 0.40 ± 0.14 | 0.83 ± 0.12 | 0.75 ± 0.07 | 0.81 ± 0.09 |
| StandardMaze | 0.28 ± 0.11 | 0.99 ± 0.01 | **1.00 ± 0.00** | 0.98 ± 0.02 | 0.90 ± 0.05 |
| StandardMaze2 | 0.30 ± 0.10 | 0.85 ± 0.06 | 0.95 ± 0.05 | 0.84 ± 0.12 | **1.00 ± 0.00** |
| StandardMaze3 | 0.56 ± 0.14 | 0.93 ± 0.04 | **1.00 ± 0.00** | 0.98 ± 0.02 | **1.00 ± 0.00** |
| Mean | 0.34 ± 0.05 | 0.76 ± 0.05 | 0.88 ± 0.04 | 0.83 ± 0.03 | 0.86 ± 0.02 |

Table 5: Minigrid Solve Rate (2)

| Level | PLR - PVL | ACCEL - PVL | PLR - MNA | ACCEL - MNA | DEGen - MNA |
|---|---|---|---|---|---|
| SixteenRooms | **1.00 ± 0.00** | **1.00 ± 0.00** | **1.00 ± 0.00** | **1.00 ± 0.00** | **1.00 ± 0.00** |
| SixteenRooms2 | 0.95 ± 0.04 | 0.88 ± 0.05 | **1.00 ± 0.00** | **1.00 ± 0.00** | **1.00 ± 0.00** |
| Labyrinth | 0.71 ± 0.09 | 0.86 ± 0.04 | 0.88 ± 0.06 | 0.68 ± 0.09 | **0.98 ± 0.02** |
| LabyrinthFlipped | 0.63 ± 0.11 | 0.78 ± 0.07 | 0.73 ± 0.11 | 0.73 ± 0.08 | **0.91 ± 0.09** |
| Labyrinth2 | 0.51 ± 0.07 | 0.73 ± 0.08 | 0.93 ± 0.06 | **0.95 ± 0.02** | 0.79 ± 0.12 |
| StandardMaze | 0.63 ± 0.09 | 0.84 ± 0.06 | 0.98 ± 0.02 | **1.00 ± 0.00** | **1.00 ± 0.00** |
| StandardMaze2 | 0.63 ± 0.10 | 0.88 ± 0.07 | 0.86 ± 0.08 | 0.99 ± 0.01 | **1.00 ± 0.00** |
| StandardMaze3 | 0.91 ± 0.05 | 0.99 ± 0.01 | **1.00 ± 0.00** | 0.98 ± 0.02 | **1.00 ± 0.00** |
| Mean | 0.75 ± 0.03 | 0.87 ± 0.03 | 0.92 ± 0.02 | 0.91 ± 0.01 | **0.96 ± 0.03** |

**Key Minigrid 13x13 Results**

Table 6: Key Minigrid 13x13 Solve Rate (1)

| Level | Initial Gen - MNA | DR | SFL | PLR - MaxMC | ACCEL - MaxMC |
|---|---|---|---|---|---|
| SixteenRooms_Key | 0.18 ± 0.12 | 0.59 ± 0.12 | 0.58 ± 0.12 | 0.41 ± 0.11 | 0.83 ± 0.04 |
| SixteenRooms2_Key | 0.12 ± 0.12 | 0.55 ± 0.13 | 0.83 ± 0.10 | 0.19 ± 0.05 | 0.83 ± 0.08 |
| FourRooms_Key | 0.03 ± 0.02 | 0.55 ± 0.13 | **0.98 ± 0.02** | 0.10 ± 0.08 | 0.96 ± 0.02 |
| FourRooms2_Key | 0.05 ± 0.04 | 0.88 ± 0.06 | 0.93 ± 0.07 | 0.79 ± 0.10 | 0.98 ± 0.02 |
| FourRooms3_Key | 0.29 ± 0.14 | 0.51 ± 0.14 | 0.85 ± 0.06 | 0.10 ± 0.06 | 0.95 ± 0.05 |
| StandardMaze_Key | 0.01 ± 0.01 | 0.09 ± 0.04 | 0.35 ± 0.12 | 0.24 ± 0.07 | 0.20 ± 0.10 |
| StandardMaze2_Key | 0.00 ± 0.00 | 0.61 ± 0.13 | 0.60 ± 0.14 | 0.40 ± 0.11 | 0.26 ± 0.07 |
| StandardMaze3_Key | 0.00 ± 0.00 | 0.36 ± 0.14 | 0.30 ± 0.14 | 0.09 ± 0.06 | 0.75 ± 0.05 |
| Mean | 0.08 ± 0.04 | 0.52 ± 0.05 | 0.68 ± 0.04 | 0.29 ± 0.03 | 0.72 ± 0.02 |

Table 7: Key Minigrid 13x13 Solve Rate (2)

| Level | PLR - PVL | ACCEL - PVL | PLR - MNA | ACCEL - MNA | DEGen - MNA |
|---|---|---|---|---|---|
| SixteenRooms_Key | 0.15 ± 0.10 | 0.55 ± 0.04 | 0.95 ± 0.03 | 0.99 ± 0.01 | **1.00 ± 0.00** |
| SixteenRooms2_Key | 0.00 ± 0.00 | 0.63 ± 0.10 | 0.90 ± 0.06 | 0.88 ± 0.10 | **1.00 ± 0.00** |
| FourRooms_Key | 0.00 ± 0.00 | 0.60 ± 0.12 | 0.95 ± 0.04 | **0.98 ± 0.02** | 0.94 ± 0.05 |
| FourRooms2_Key | 0.06 ± 0.03 | 0.80 ± 0.06 | **1.00 ± 0.00** | 0.99 ± 0.01 | **1.00 ± 0.00** |
| FourRooms3_Key | 0.00 ± 0.00 | 0.66 ± 0.09 | 0.95 ± 0.03 | 0.98 ± 0.02 | **1.00 ± 0.00** |
| StandardMaze_Key | 0.00 ± 0.00 | 0.00 ± 0.00 | 0.46 ± 0.10 | 0.85 ± 0.08 | **0.93 ± 0.06** |
| StandardMaze2_Key | 0.00 ± 0.00 | 0.06 ± 0.02 | 0.73 ± 0.11 | **0.75 ± 0.08** | 0.63 ± 0.14 |
| StandardMaze3_Key | 0.00 ± 0.00 | 0.30 ± 0.09 | 0.78 ± 0.07 | 0.84 ± 0.09 | **0.94 ± 0.05** |
| Mean | 0.03 ± 0.01 | 0.45 ± 0.01 | 0.84 ± 0.01 | 0.90 ± 0.03 | **0.93 ± 0.02** |

**Key Minigrid 17x17 Results**

Table 8: Key Minigrid 17x17 Solve Rate

| Level | PLR - MNA | ACCEL - MNA | DEGen - MNA |
|---|---|---|---|
| SixteenRooms_Key | 0.99 ± 0.01 | 0.86 ± 0.05 | **1.00 ± 0.00** |
| SixteenRooms2_Key | 0.86 ± 0.03 | 0.40 ± 0.13 | **0.95 ± 0.05** |
| FourRooms_Key | 0.80 ± 0.11 | 0.64 ± 0.09 | **0.94 ± 0.05** |
| FourRooms2_Key | 0.92 ± 0.05 | 0.81 ± 0.10 | **0.95 ± 0.04** |
| FourRooms3_Key | 0.88 ± 0.07 | 0.65 ± 0.09 | **0.94 ± 0.04** |
| StandardMaze_Key | **0.69 ± 0.13** | 0.19 ± 0.08 | 0.23 ± 0.05 |
| StandardMaze2_Key | 0.29 ± 0.09 | 0.21 ± 0.07 | **0.50 ± 0.07** |
| StandardMaze3_Key | 0.73 ± 0.09 | **0.83 ± 0.06** | 0.73 ± 0.09 |
| Mean | 0.77 ± 0.04 | 0.57 ± 0.03 | **0.78 ± 0.03** |

**Key Minigrid 21x21 Results**

Table 9: Key Minigrid 21x21 Solve Rate

| Level | PLR - MNA | ACCEL - MNA | DEGen - MNA |
|---|---|---|---|
| SixteenRooms_Key | 0.60 ± 0.11 | 0.60 ± 0.12 | **1.00 ± 0.00** |
| SixteenRooms2_Key | 0.40 ± 0.09 | 0.45 ± 0.11 | **0.99 ± 0.01** |
| FourRooms_Key | 0.21 ± 0.10 | 0.41 ± 0.11 | **0.99 ± 0.01** |
| FourRooms2_Key | **0.93 ± 0.04** | 0.59 ± 0.09 | **0.93 ± 0.05** |
| FourRooms3_Key | 0.35 ± 0.12 | 0.19 ± 0.08 | **0.98 ± 0.02** |
| StandardMaze_Key | 0.34 ± 0.11 | 0.06 ± 0.03 | **0.56 ± 0.11** |
| StandardMaze2_Key | 0.26 ± 0.12 | 0.00 ± 0.00 | **0.39 ± 0.12** |
| StandardMaze3_Key | 0.38 ± 0.11 | 0.16 ± 0.07 | **0.61 ± 0.11** |
| Mean | 0.43 ± 0.05 | 0.31 ± 0.04 | **0.80 ± 0.03** |

# C Sokoban Environment

We have additionally performed experiments in a Sokoban-style environment. As in standard Sokoban, the agent aims to get all boxes to their storage locations. The agent receives a sparse reward when completing a level inversly proportional to how many steps were required to complete the level. For this domain, similarly to minigrid the agent has 5x5 observation space ahead of the agent, and an action space of move *forward*, turn *left* or turn *right*. The agent also has a *reset* action that resets the agent and boxes to their starting locations.

We used 9x9 levels for training - for DR, PLR and SFL, the random level generator generated levels that had 15 walls, and between 1 and 10 boxes. For DEGen, the generator could fill each newly observed cell in the environment with an empty square/wall/box/storage location/box on storage location. For all methods, we used identical hyperparameters to the minigrid environment (See Table 1).

## C.1 Sokoban Results

From the results shown in Figure 11 and Tables 10 and 11, we see a large range in the performance of the various UED methods. We see that ACCEL using MaxMC is the best performing method, although comparable performance is achieved using DEGen. In Sokoban, we see that MaxMC outperforms MNA in the replay-based methods, with a substantial performance difference when used with ACCEL. Sokoban presents unique challenges compared to the previous environments used in this work. Primarily, the majority of randomly-sampled levels tend to be impossible, given that it only takes one box or storage location to be unreachable to make the entire level unsolvable. However, it is also likely that a high proportion of the solvable randomly-sampled levels will be difficult.

As all impossible levels will necessarily score zero with both MaxMC and MNA, We hypothesise that MaxMC may be a better metric than MNA in domains where difficult levels represent a higher proportion of the non-zero scoring randomly-sampled levels. The high performance of ACCEL in Sokoban is likely due to the clear difficulty scaling that can be achieved with ACCEL-like level evolution. In our implementation of ACCEL, one of the possible mutations is to add or remove a box/storage-location pair. This enables gradual difficulty evolution in Sokoban, which is less likely in the minigrid enviroments. As such, ACCEL is highly effective at generating an curriculum.

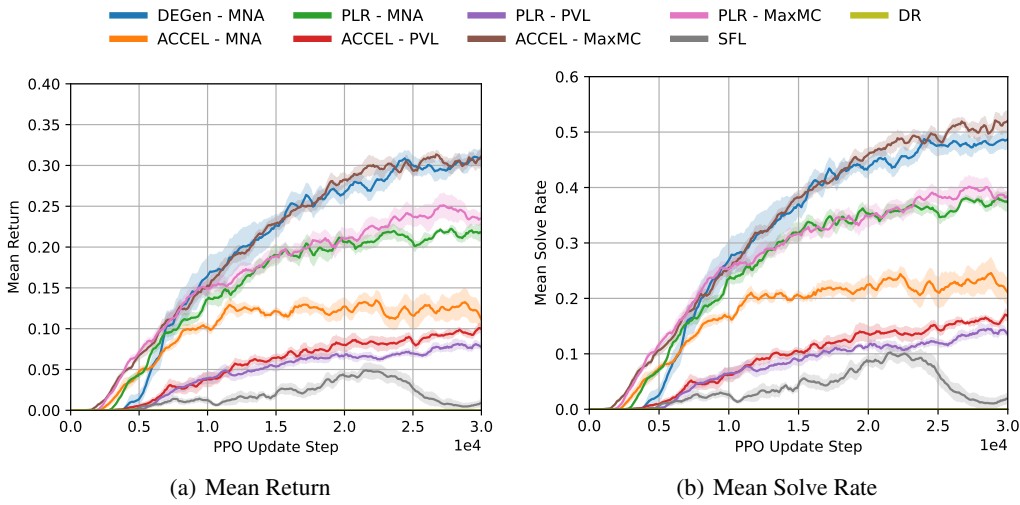

(a) Mean Return  (b) Mean Solve Rate

Figure 11: Sokoban zero-shot performance on hand-designed test set, showing mean and standard error across 8 runs.

We do however see in Tables 10 and 11 that a number of levels, those marked in *italics*, are not solved by any method. This suggests that there is room for future work to enable zero-shot performance on more difficult levels, and that Sokoban may be an interesting environment for future UED research.

## C.2 Sokoban Zero-shot Transfer Levels

For the zero-shot transfer set, we have used the first 20 Sokoban Jr levels that do not exceed 13x13 in size.

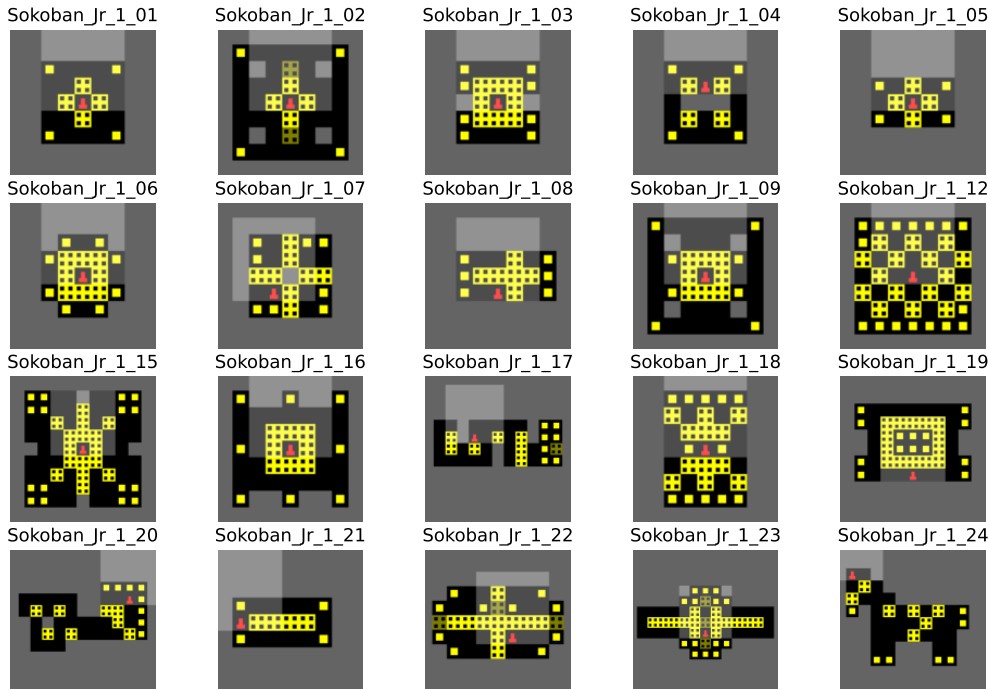

Figure 12: Hand designed evaluation levels for sokoban

## C.3 Sokoban Tabular Results

Table 10: Sokoban Solve Rate (1)

| Level | DR | SFL | PLR - MaxMC | ACCEL - MaxMC |
|---|---|---|---|---|
| Sokoban_Jr_1_01 | 0.00 ± 0.00 | 0.23 ± 0.15 | **1.00 ± 0.00** | **1.00 ± 0.00** |
| Sokoban_Jr_1_02 | 0.00 ± 0.00 | 0.18 ± 0.12 | 0.94 ± 0.05 | 0.98 ± 0.02 |
| Sokoban_Jr_1_03 | 0.00 ± 0.00 | 0.00 ± 0.00 | 0.46 ± 0.13 | **0.83 ± 0.12** |
| Sokoban_Jr_1_04 | 0.00 ± 0.00 | 0.00 ± 0.00 | 0.16 ± 0.07 | 0.30 ± 0.09 |
| Sokoban_Jr_1_05 | 0.00 ± 0.00 | 0.00 ± 0.00 | 0.42 ± 0.14 | 0.45 ± 0.12 |
| Sokoban_Jr_1_06 | 0.00 ± 0.00 | 0.00 ± 0.00 | 0.74 ± 0.11 | 0.75 ± 0.11 |
| Sokoban_Jr_1_07 | 0.00 ± 0.00 | 0.00 ± 0.00 | 0.60 ± 0.14 | 0.96 ± 0.03 |
| Sokoban_Jr_1_08 | 0.00 ± 0.00 | 0.00 ± 0.00 | 0.21 ± 0.09 | **0.50 ± 0.13** |
| Sokoban_Jr_1_09 | 0.00 ± 0.00 | 0.00 ± 0.00 | 0.65 ± 0.05 | 0.81 ± 0.04 |
| Sokoban_Jr_1_12 | 0.00 ± 0.00 | 0.00 ± 0.00 | 0.00 ± 0.00 | 0.29 ± 0.06 |
| *Sokoban_Jr_1_15* | *0.00 ± 0.00* | *0.00 ± 0.00* | *0.00 ± 0.00* | *0.00 ± 0.00* |
| Sokoban_Jr_1_16 | 0.00 ± 0.00 | 0.00 ± 0.00 | 0.93 ± 0.04 | **0.96 ± 0.02** |
| *Sokoban_Jr_1_17* | *0.00 ± 0.00* | *0.00 ± 0.00* | *0.00 ± 0.00* | *0.00 ± 0.00* |
| Sokoban_Jr_1_18 | 0.00 ± 0.00 | 0.00 ± 0.00 | 0.04 ± 0.03 | 0.46 ± 0.10 |
| Sokoban_Jr_1_19 | 0.00 ± 0.00 | 0.00 ± 0.00 | 0.00 ± 0.00 | **0.23 ± 0.11** |
| *Sokoban_Jr_1_20* | *0.00 ± 0.00* | *0.00 ± 0.00* | *0.00 ± 0.00* | *0.00 ± 0.00* |
| Sokoban_Jr_1_21 | 0.00 ± 0.00 | 0.00 ± 0.00 | 0.96 ± 0.03 | **1.00 ± 0.00** |
| Sokoban_Jr_1_22 | 0.00 ± 0.00 | 0.00 ± 0.00 | 0.14 ± 0.02 | **0.64 ± 0.10** |
| *Sokoban_Jr_1_23* | *0.00 ± 0.00* | *0.00 ± 0.00* | *0.00 ± 0.00* | *0.00 ± 0.00* |
| Sokoban_Jr_1_24 | 0.00 ± 0.00 | 0.00 ± 0.00 | 0.04 ± 0.03 | **0.14 ± 0.05** |
| Mean | 0.00 ± 0.00 | 0.02 ± 0.01 | 0.36 ± 0.01 | **0.51 ± 0.02** |

Table 11: Sokoban Solve Rate (2)

| Level | PLR - PVL | ACCEL - PVL | PLR - MNA | ACCEL - MNA | DEGen - MNA |
|---|---|---|---|---|---|
| Sokoban_Jr_1_01 | **1.00 ± 0.00** | 0.99 ± 0.01 | 0.98 ± 0.02 | 0.89 ± 0.11 | **1.00 ± 0.00** |
| Sokoban_Jr_1_02 | 0.24 ± 0.05 | 0.46 ± 0.06 | 0.98 ± 0.02 | 0.49 ± 0.07 | **1.00 ± 0.00** |
| Sokoban_Jr_1_03 | 0.09 ± 0.07 | 0.25 ± 0.10 | 0.71 ± 0.11 | 0.15 ± 0.08 | 0.49 ± 0.14 |
| Sokoban_Jr_1_04 | 0.00 ± 0.00 | 0.00 ± 0.00 | **0.44 ± 0.09** | 0.14 ± 0.08 | 0.35 ± 0.09 |
| Sokoban_Jr_1_05 | 0.10 ± 0.10 | 0.01 ± 0.01 | **0.58 ± 0.12** | 0.48 ± 0.12 | 0.24 ± 0.16 |
| Sokoban_Jr_1_06 | 0.05 ± 0.03 | 0.03 ± 0.02 | 0.50 ± 0.14 | 0.46 ± 0.12 | **0.94 ± 0.03** |
| Sokoban_Jr_1_07 | 0.19 ± 0.13 | 0.51 ± 0.16 | 0.35 ± 0.09 | 0.19 ± 0.07 | **1.00 ± 0.00** |
| Sokoban_Jr_1_08 | 0.00 ± 0.00 | 0.00 ± 0.00 | 0.15 ± 0.11 | 0.00 ± 0.00 | **0.50 ± 0.15** |
| Sokoban_Jr_1_09 | 0.14 ± 0.04 | 0.13 ± 0.07 | 0.68 ± 0.06 | 0.18 ± 0.06 | **0.85 ± 0.06** |
| Sokoban_Jr_1_12 | 0.00 ± 0.00 | 0.00 ± 0.00 | 0.00 ± 0.00 | 0.00 ± 0.00 | **0.30 ± 0.15** |
| *Sokoban_Jr_1_15* | *0.00 ± 0.00* | *0.00 ± 0.00* | *0.00 ± 0.00* | *0.00 ± 0.00* | *0.00 ± 0.00* |
| Sokoban_Jr_1_16 | 0.25 ± 0.05 | 0.13 ± 0.06 | 0.95 ± 0.03 | 0.60 ± 0.08 | 0.95 ± 0.03 |
| *Sokoban_Jr_1_17* | *0.00 ± 0.00* | *0.00 ± 0.00* | *0.00 ± 0.00* | *0.00 ± 0.00* | *0.00 ± 0.00* |
| Sokoban_Jr_1_18 | 0.00 ± 0.00 | 0.03 ± 0.02 | 0.00 ± 0.00 | 0.00 ± 0.00 | **0.69 ± 0.14** |
| Sokoban_Jr_1_19 | 0.00 ± 0.00 | 0.00 ± 0.00 | 0.00 ± 0.00 | 0.00 ± 0.00 | 0.10 ± 0.06 |
| *Sokoban_Jr_1_20* | *0.00 ± 0.00* | *0.00 ± 0.00* | *0.00 ± 0.00* | *0.00 ± 0.00* | *0.00 ± 0.00* |
| Sokoban_Jr_1_21 | 0.51 ± 0.09 | 0.56 ± 0.12 | 0.76 ± 0.09 | 0.68 ± 0.12 | 0.99 ± 0.01 |
| Sokoban_Jr_1_22 | 0.00 ± 0.00 | 0.06 ± 0.05 | 0.09 ± 0.05 | 0.00 ± 0.00 | 0.30 ± 0.11 |
| *Sokoban_Jr_1_23* | *0.00 ± 0.00* | *0.00 ± 0.00* | *0.00 ± 0.00* | *0.00 ± 0.00* | *0.00 ± 0.00* |
| Sokoban_Jr_1_24 | 0.00 ± 0.00 | 0.01 ± 0.01 | 0.05 ± 0.04 | 0.00 ± 0.00 | 0.09 ± 0.03 |
| Mean | 0.13 ± 0.01 | 0.16 ± 0.01 | 0.36 ± 0.01 | 0.21 ± 0.02 | 0.49 ± 0.02 |

# D Additional Results

## D.1 MNA and Existing Methods

To directly examine the effectiveness of MNA compared to existing regret metics, we show zero-shot performance of ACCEL and PLR using MNA, PVL and MaxMC.

### Minigrid

In Figures 13 and 14, we illustrate the relative performance of each of these metrics in the standard minigrid domain and show that MNA clearly outperforms other metrics, whether using either PLR or ACCEL.

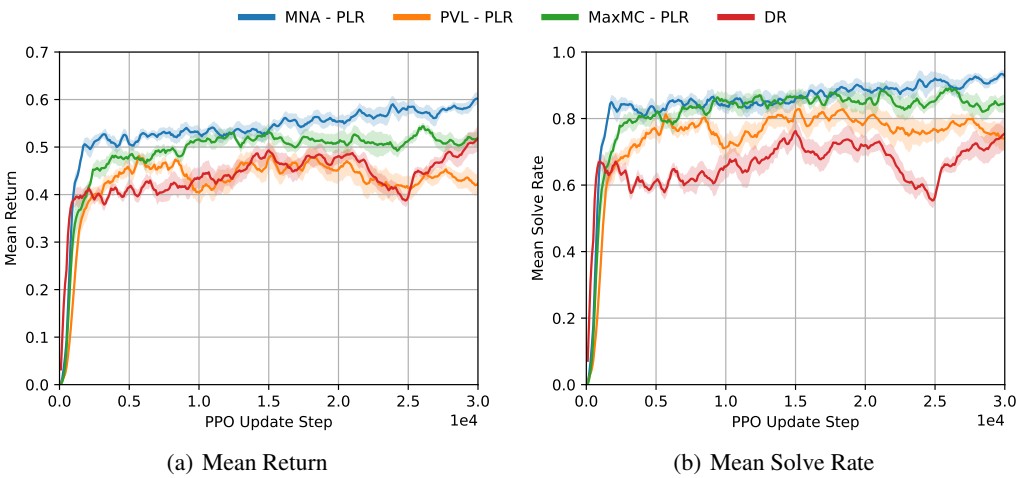

(a) Mean Return

(b) Mean Solve Rate

Figure 13: Minigrid - comparison of PLR performance using different metrics

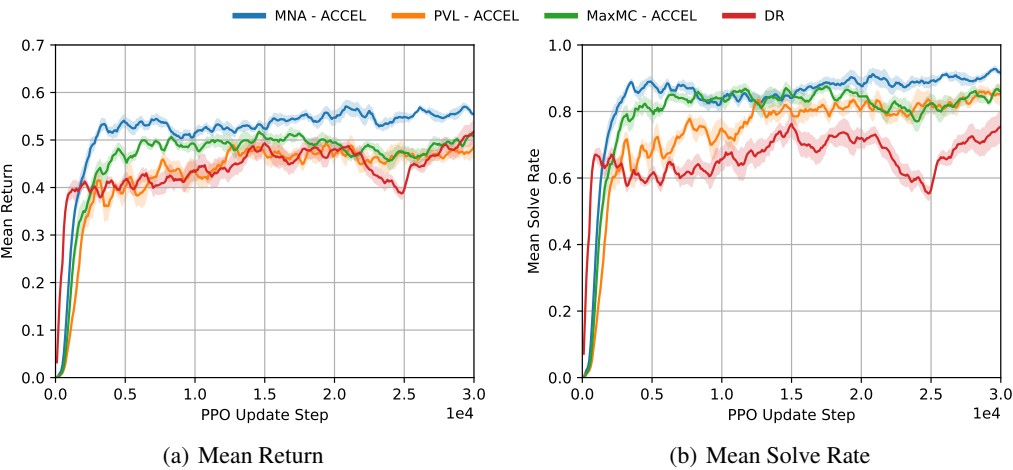

(a) Mean Return

(b) Mean Solve Rate

Figure 14: Minigrid - comparison of ACCEL performance using different metrics

**Key Minigrid**

In Figures 15 and 16, we compare the same methods but on the key minigrid domain instead. Here, we again see that MNA outperforms existing methods - including a substantial performance improvement when using PLR.

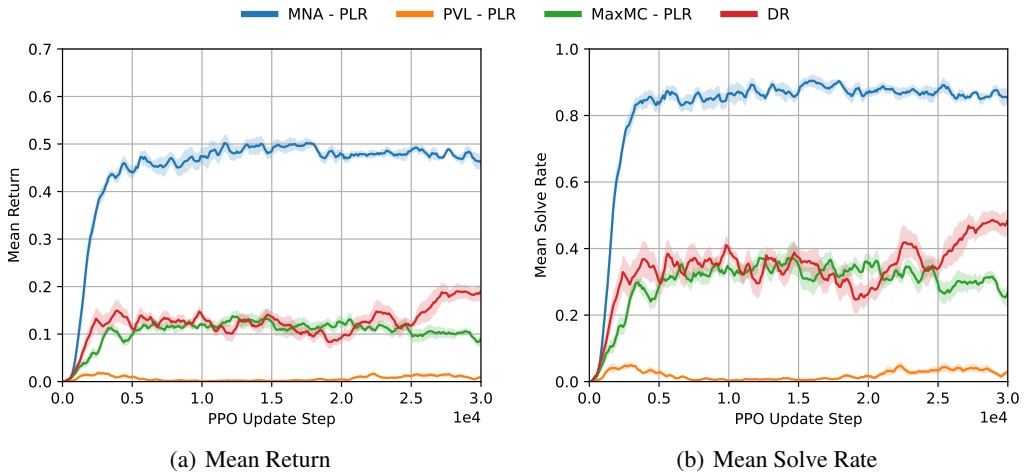

(a) Mean Return   (b) Mean Solve Rate

Figure 15: Key Minigrid - comparison of PLR performance using different metrics

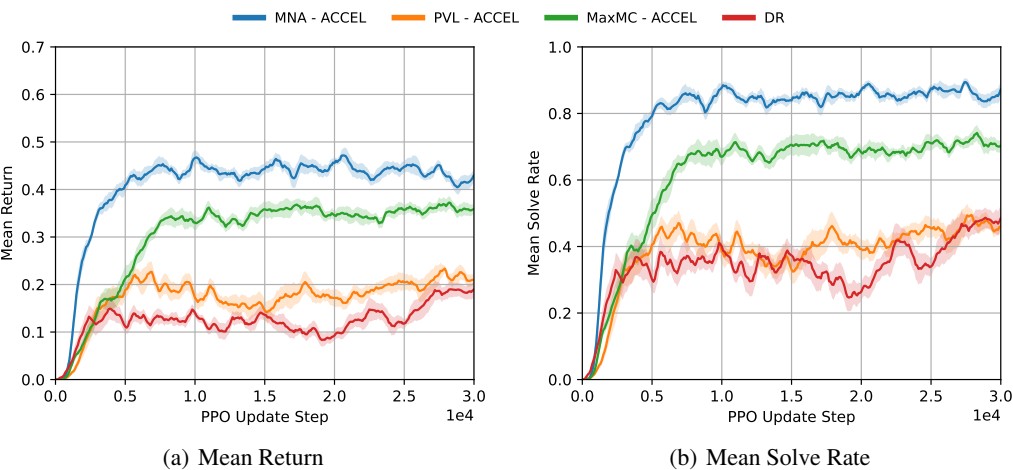

(a) Mean Return   (b) Mean Solve Rate

Figure 16: Key Minigrid - comparison of ACCEL performance using different metrics

## D.2 DEGen and Existing Regret Approximations

In order to illustrate the ineffectiveness of existing regret approximations when used as optimisation objectives for training a teacher, we show the relative performance of DEGen using MNA, PVL and MaxMC. Figures 17 and 18 show that MNA consistently outperforms PVL and MaxMC. We also see here that using a teacher trained using PVL and MaxMC results in at best equivalent, but generally worse, performance compared to naive domain randomisation.

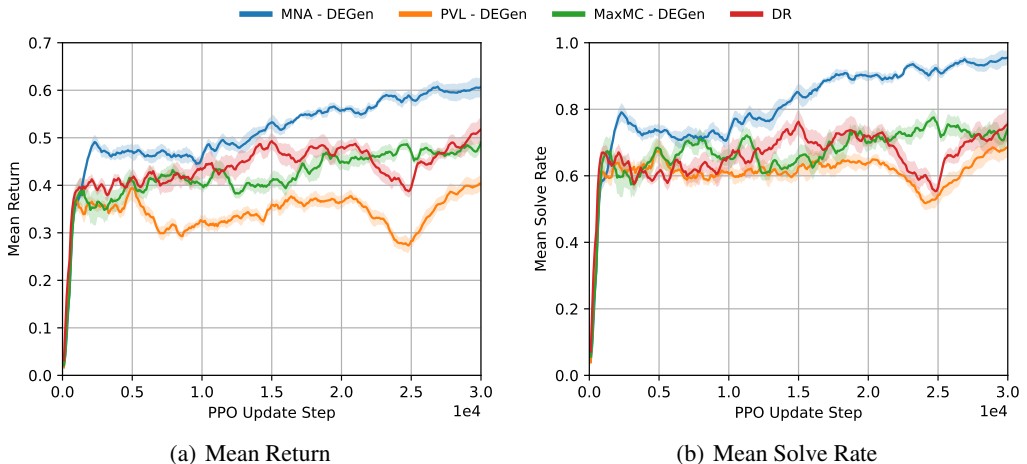

(a) Mean Return       (b) Mean Solve Rate

Figure 17: Minigrid - comparison of DEGen performance using different metrics

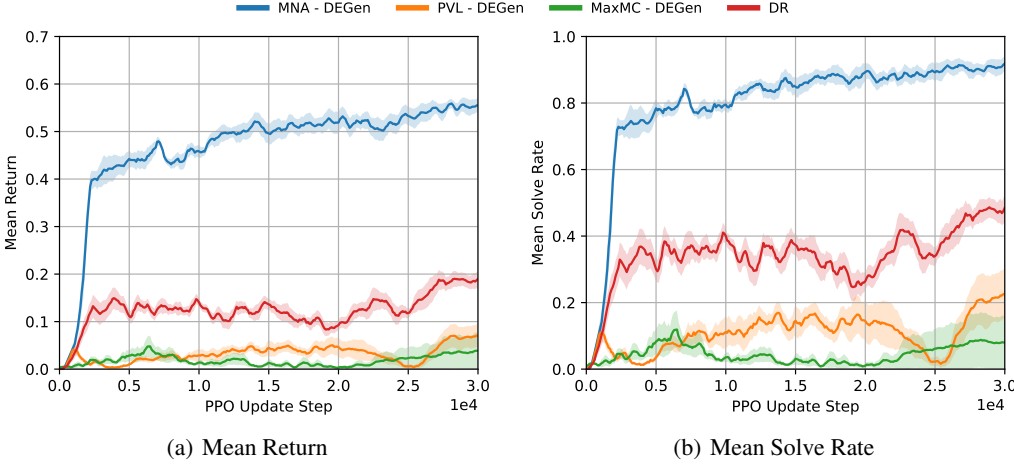

(a) Mean Return       (b) Mean Solve Rate

Figure 18: Key Minigrid - comparison of DEGen performance using different metrics

## D.3 DEGen vs Initial Gen

In Figures 19 and 20, we show the performance of DEGen compared to the performance of a generator that generates the full level prior to student rollouts. We include both a standard level generator *Initial Gen*, identical to the PAIRED generator in JaxUED [10], as well as *Initial Gen (Rand)*, which randomly places the agent in the level before the rest of the level is constructed. We see that both methods performance worse than domain randomisation.

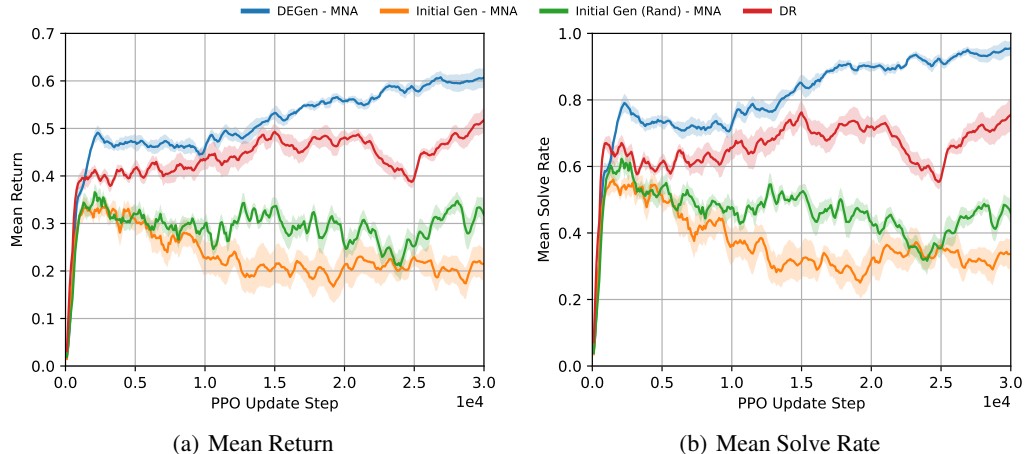

(a) Mean Return

(b) Mean Solve Rate

Figure 19: Minigrid - comparison of Initial Gen and DEGen

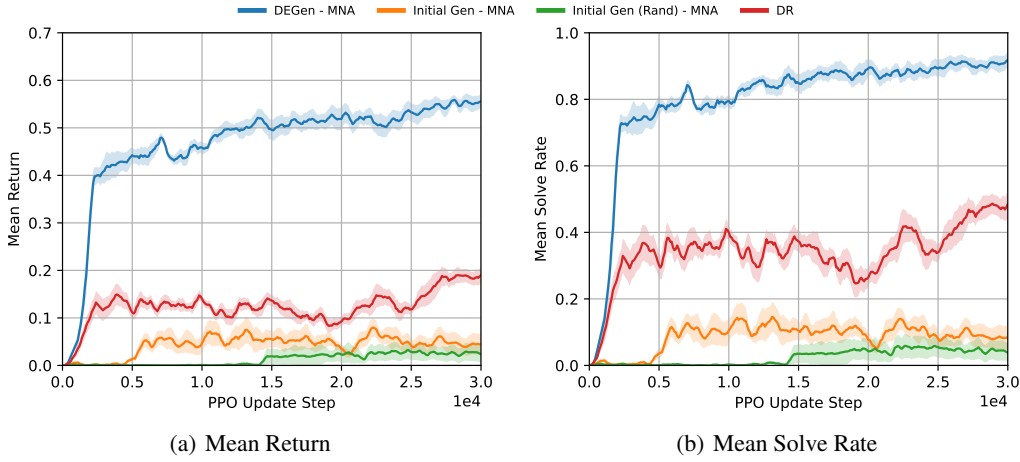

(a) Mean Return

(b) Mean Solve Rate

Figure 20: Key Minigrid - comparison of Initial Gen and DEGen

## D.4 PAIRED

Finally, we examine the performance of DEGen compared to PAIRED [11]. In standard minigrid, we see that PAIRED performs very similarly to domain randomisation, and worse than DEGen. Additionally, we see in the key minigrid domain the limitations of the PAIRED regret approximation. As PAIRED relies on the antagonist's performance to approximate the best possible level return, high scoring levels require the antagonist to perform well. However, if a level is challenging due to some obstacle the student has not previously encountered, it is likely that the antagonist will also perform poorly, given it has been trained on the same set of levels as the student. Therefore, the PAIRED generator is unlikely to generate levels requiring the antagonist to use the key, and so as the student agent has not encountered levels similar to the zero-shot hand-designed levels that require the key, zero-shot performance is extremely poor.

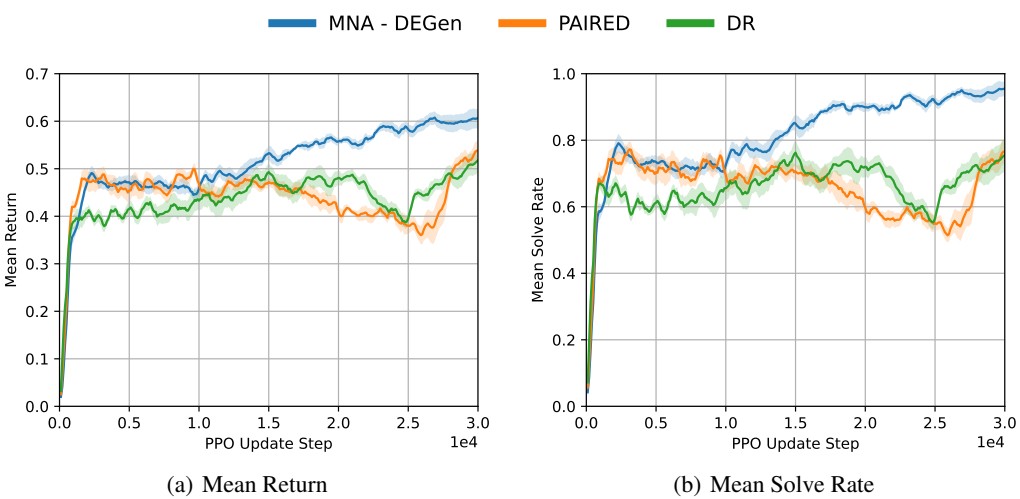

(a) Mean Return

(b) Mean Solve Rate

Figure 21: Minigrid - comparison of PAIRED and DEGen

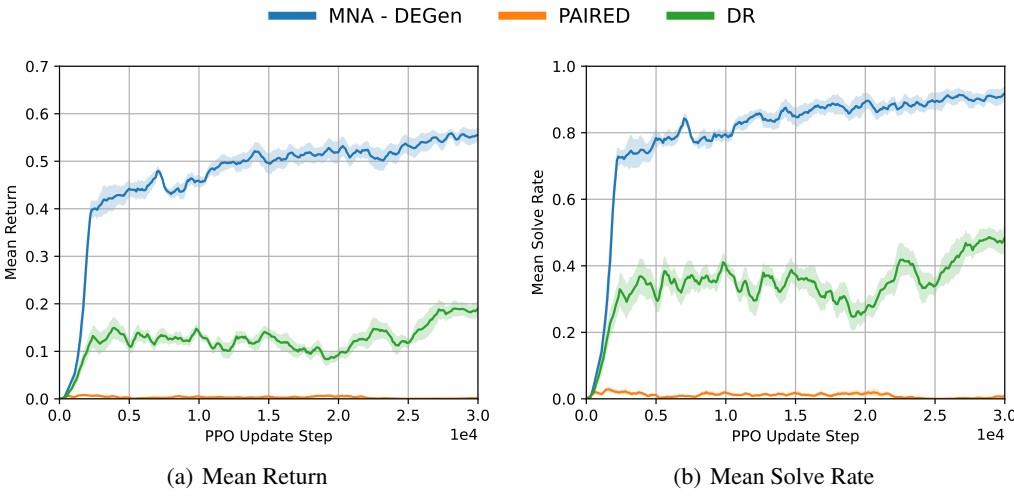

(a) Mean Return

(b) Mean Solve Rate

Figure 22: Key Minigrid - comparison of PAIRED and DEGen

## D.5 Training Level Examples

We have included examples of levels generated by each method in the repository at `https://github.com/HarryMJMead/Dynamic-Environment-Generation-for-UED`. These levels were all sampled from the final training step.

