# OpenReview forum: "Improving Regret Approximation for Unsupervised Dynamic Environment Generation"
_NeurIPS.cc/2025/Conference — NeurIPS 2025 poster_

### Official Review · Reviewer_6nip · 2025-06-17

**Clarity:** 3
**Significance:** 2
**Originality:** 2
**Rating:** 4
**Confidence:** 4

**Summary:**

This paper proposes Dynamic Environment Generation for Unsupervised Environment Design with a new regret approximation. This method outperforms existing methods in some cases.

**Questions:**

- It is confusing that the **zero-shot performance** has rewards curves with the updating steps of parameters. In many RL works that focus on generalization and transferring, zero-shot performance will be shown with success rates in unseen new tasks without updating any parameters of the whole agent. For instance, in the citation [8] of this work, they utilize 'percent successful trials in environments used to test zero-shot transfer'.
- The method DEG may have good consistency because it utilizes a mask to build partial observation environments for simulating the change and dynamics in the environments. However, this method will lose diversity. In environment generation, consistency causes overfitting while diversity causes instability. This paper requires further discussion on the effectiveness of its method.
- Many of the test experiments are hand-designed. It will undermine the credibility of the results.
- In both the appendix and the main text, the results seem to diverge from the claim in the paper. Because the performance differences between the baselines are minimal in minigrid but are quite different in key-minigrid. The task of key-minigrid has bottleneck states and requires strong exploration capabilities. The outperformance owing to that the curricula built by partial observation can give guidance to encourage to explore, not the intrinsic advantage of the method itself. These are not suitable comparisons.

**Ethical Concerns:**

["NO or VERY MINOR ethics concerns only"]

**Final Justification:**

The author has addressed most of my concerns satisfactorily. I will raise the score to ‘Borderline Accept’.

**Limitations:**

Yes but not sufficient.

**Quality:**

2

**Strengths And Weaknesses:**

Strengths:
- The focused problem is important and this paper demonstrates good clarity of expression.

Weakness:
- This paper lacks sufficient analyses of the effectiveness of the method.
- Parts of the setup in the experiments are confused.

---

> ### Author Rebuttal · Authors · 2025-07-31
>
> # Questions
>
> ***It is confusing that the zero-shot performance has rewards curves …***
>
> To provide further clarity, the zero-shot return plots are showing the return and solve rate of the agent on the unseen test set as the agent is trained using a specific UED method. The agent is not being trained on the unseen levels - it is being trained using the levels generated/curated by the UED method. Plots showing zero-shot performance throughout training are standard in UED literature [1,2,3,4,5]
>
> ***The method DEG may have good consistency because it utilizes a mask to build partial observation …***
>
> We found that introducing some randomness to the environment initialisation - as outlined in Section 3 of the paper - was sufficient to ensure level diversity throughout training. We did not observe any instability induced by this diversity. We would appreciate a follow-up question if this does not fully address your concerns.
>
> ***Many of the test experiments are hand-designed …***
>
> Whilst there could be issues with using hand-designed test sets, it is the standard practice in UED literature [1,2,3,4,5] to show zero-shot performance.
>
> ***In both the appendix and the main text, the results seem to diverge from the claim in the paper …***
>
> We disagree with our results diverging from claims of the paper. We agree that the key-minigrid task does require strong exploration capabilities but our results show that MNA and DEGen are capable of facilitating a curriculum that enables the agent to learn this exploration, given they both outperform existing baselines. As you suggest, “The outperformance owing to that the curricula built by partial observation can give guidance to encourage to explore”, but this is an intrinsic advantage of the method. We would appreciate a follow-up question if you have further issues with this not being a suitable comparison.
>
>
> [1] Chung, Hojun, et al. "Adversarial environment design via regret-guided diffusion models." Advances in Neural Information Processing Systems 37 (2024): 63715-63746.
>
> [2] Parker-Holder, Jack, et al. "Evolving curricula with regret-based environment design." International Conference on Machine Learning. PMLR, 2022.
>
> [3] Rutherford, Alexander, et al. "No regrets: Investigating and improving regret approximations for curriculum discovery." Advances in Neural Information Processing Systems 37 (2024): 16071-16101.
>
> [4] Jiang, Minqi, et al. "Replay-guided adversarial environment design." Advances in Neural Information Processing Systems 34 (2021): 1884-1897.
>
> [5] Mediratta, Ishita, et al. "Stabilizing unsupervised environment design with a learned adversary." Conference on Lifelong Learning Agents. PMLR, 2023.

---

> > ### Comment · Reviewer_6nip · 2025-08-05
> >
> > The author has addressed most of my concerns satisfactorily. I will raise the score to ‘Borderline Accept’.

---

### Official Review · Reviewer_D6YH · 2025-07-03

**Clarity:** 3
**Significance:** 3
**Originality:** 3
**Rating:** 5
**Confidence:** 4

**Summary:**

This paper proposes to improve unsupervised environment generation methods. It does this using two ideas: 1) to avoid the level generator struggles with credit assignment just have it generate the level around the agent and 2) a new approximation for regret. It shows that as UED as scaled to larger environments, this method begins to outperform the baselines.

**Questions:**

- In line 187, why does decreasing the horizon of this V function increase the likelihood of conservatism? Is it purely that you're decreasing the number of random variables you're taking a max over?
- Why does figure 5 use return instead of solve rate?
- Did you list your hyperparameter tuning procedure for all methods in the paper somewhere? Did I just miss it?

**Ethical Concerns:**

["NO or VERY MINOR ethics concerns only"]

**Final Justification:**

My original score was provided based on not observing any significant errors and the results seeming to be a significant improvement. The authors answered my questions so I maintain my score.

**Limitations:**

yes.

**Quality:**

3

**Strengths And Weaknesses:**

I thought this was a really good paper. The ideas are clear and understandable, the results seem to be significant and the methods can be straightforwardly implemented. They test against standard, strong baselines. I have very little to say because I think the paper is very solid.

### Serious Weaknesses
- Please put your hyperparameter tuning procedure for your algorithms and the baselines into the supplement or paper. Without this it is hard to tell if the baselines were really afforded their due in tackling the problems under consideration.

### Less serious weaknesses / suggestions
- It probably would make the paper stronger to study more environments besides minigrid as it would help make the case that the technique is more general. It won't change my score, it's just a suggestion.


### Nitpicks (you don't necessarily have to follow them and it won't change my score but I think you should make these changes to make your paper better)
- Going from equation 7 to equation 8 is a bit of a leap, you could hold the reader's hand more here.
- I think in 178 you should be clearer that this is the true value function

---

> ### Author Rebuttal · Authors · 2025-07-31
>
> Thank you for your review. We very much appreciate your positive response to our work.
>
> ## Hyperparameter Tuning
>
> For hyperparameter tuning, we have relied on previous work [1] using the same implementations for our baselines. Full details are shown in the Appendix of [1], and we will state this in the final version of the paper
>
> ## Additional Environments
>
> Reviewer 7GXd suggested some additional environments that may be suitable. We have done some preliminary experiments in Sokoban with promising initial results, and so may be able to include these in a final version of the paper.
>
> # Questions
>
> ***In line 187, why does decreasing the horizon of this V function increase the likelihood of conservatism …***
>
> Reducing the horizon results in a reduced number of different value approximations for the state (see Eqn 7), and so the greater likelihood of conservatism is just to do with a reduced number of variables over which you are taking a max over.
>
> ***Why does figure 5 use return instead of solve rate?***
>
> We have only included one of return and solve rate in Figure 5 for the sake of space and chose return because it not only reflects the proportion of times the agent solves the level but also how quickly the level is solved. However, we do acknowledge that it may be less interpretable to the reader so we will include the solve rate plots in the Appendix, but qualitatively, these solve rate plots show the same patterns as seen in the return plots.
>
>
> [1] Rutherford, Alexander, et al. "No regrets: Investigating and improving regret approximations for curriculum discovery." Advances in Neural Information Processing Systems 37 (2024): 16071-16101.

---

### Official Review · Reviewer_dzPV · 2025-07-07

**Clarity:** 4
**Significance:** 4
**Originality:** 4
**Rating:** 5
**Confidence:** 4

**Summary:**

* Introduces (1) a new regret approximation for the UED setting, Maximized Negative Advantage (MNA), in order to better identify challenging levels; and (2) Dynamic Environment Generation for UED (DEGen) to enable UED to scale to larger environment sizes

**Questions:**

* I wonder if the DEGen idea is directly compatible with a diffusion-based generator? I.e. where each diffusion step clarifies the environment more and more; of course, this increases the complexity of training the generator, probably; but might be a more scalable approach in the long-term (assuming the initial diffusion step produces a valid level)

**Ethical Concerns:**

["NO or VERY MINOR ethics concerns only"]

**Final Justification:**

I stand by my initial review and therefore maintain my current score.

**Limitations:**

Yes

**Quality:**

4

**Strengths And Weaknesses:**

Strengths
* The paper is well-written; easy to follow; properly contextualizes related work
* Accurate (and sensitive, in the upper-difficulty region) regret approximations are indeed much-needed; a fundamental challenge with UED
* The DEGen idea is fantastically creative and novel; hadn't considered this kind of approach for improving the problem of credit assignment for UED, but is natural, and—
* The MNA and DEGen results are both convincing

Weaknesses
* It's not clear to me how general the DEGen idea is; I can imagine it being difficult to define the constraints in more realistic settings, where it is not so easy to tell which causal variables have been interacted with yet, and can be generated on the fly
* None major

---

> ### Author Rebuttal · Authors · 2025-07-31
>
> Thank you for your review. We very much appreciate your positive response to our work.
>
> We do agree that there may be some limitations with DEGen mapping the agent’s observations to the relevant environment parameters, but we believe that DEGen represents a promising starting point for future work.
>
> # Questions
>
> ***I wonder if the DEGen idea is directly compatible with a diffusion-based generator …***
>
> This definitely is an interesting approach for future work. There has already been UED work [1] using diffusion-based generation for generating whole levels, so applying this to generating partial levels seems like a natural extension. Thank you for the suggestion.
>
> [1] Chung, Hojun, et al. "Adversarial environment design via regret-guided diffusion models." Advances in Neural Information Processing Systems 37 (2024): 63715-63746.

---

### Official Review · Reviewer_7GXd · 2025-07-21

**Clarity:** 3
**Significance:** 3
**Originality:** 2
**Rating:** 5
**Confidence:** 4

**Summary:**

The authors present two methods for improving UED. First, they introduce a partial environment generation method that only produces parameters of an explored level as the agent explores it. Second, they introduce a new approximation of regret for UED that promotes more challenging levels as a regret counterpart of level replay methods that promote past challenging and diverse levels. The authors evaluate their methods on the minigrid environment and a modification that requires the agent to find a key to open a door. Both improvements yield an algorithm that outperforms standard baselines for UED on larger levels.

**Questions:**

- Some details of dynamic environment generation confuse me. Specifically, the environment generator dynamically produces a partial environment based on what the current agent observes. However, minimax regret-based methods often require a protagonist and antagonist agent. These agents could interact with different parts of the environment. How does this interact with partial environments when computing regret? Or can this method only be applied to methods with one agent?

- In the derivation of MNA, the authors assume a deterministic environment. Is this a general assumption? Why or why not? Is MNA limited to deterministic environments or could be generalized?

- The dynamic environment generation seems to be only applicable in domains where strong partial observability is present. What about fully observable domains? Is there still a benefit?

- In the paper, the minigrid levels are closely connected to observations. However, consider environments like doom, where there is significant computation to produce the observations. It seems like the inverse function of mapping from an partial observations to the free parameters needed to be filled in by the environment generation problem to be highly non-trivial. How would the authors propose adapting their method to more complex entangling of observation spaces and environment free parameters?

**Ethical Concerns:**

["NO or VERY MINOR ethics concerns only"]

**Final Justification:**

The authors did a good job of addressing my technical concerns during the rebuttal period. The original manuscript only contained one experimental domain (mini-grid), and although the core idea behind the paper is interesting, the experiments were not thorough enough  for a NeurIPS paper. The original manuscript also failed to highlight some major limitations of the proposed work. Additionally, I had some confusion on why the two techniques needed to exist in one paper.

To address these concerns, the authors added one of my suggested domains (Sokoban) and shared preliminary one-shot results. The new domain greatly strengthens the paper as Sokoban is a domain where small changes greatly affect the outcome and these changes occur quite frequently in the environment. This better supports the arguments made in the paper. The preliminary results also show a wider gap between the proposed method and baselines. I think this better justifies the proposed approach. While we didn't get to see the final results before the rebuttal period ended (which would be unreasonable), I do think the authors will be able to make a fair comparison before the camera ready period. If the paper is accepted, the authors should feel comfortable publishing negative results in this domain as it would highlight current limitations. The outcome of the additional experiments should not affect acceptance. In other words, the authors should include accurate results and not feel pressured to make their proposed method work beat the baselines.

The authors rewrote a portion of the limitations section to highlight the limitations raised by reviewers not addressed in the current manuscript. These included problems related to multiple agents, stochastic environments, and complex mappings between observations and environment representation. I think this is important because future work could address these limitations and cite this work as evidence these are open and important problems. While I would like to see a work address the complex mapping issue when doing dynamic environment generation, it is perhaps best left to future work so that the community at large can try and address it.

My main negative not addressed during the rebuttal period is lack of scholarship from the broader environment generation field. After a small literature review of procedural content generation, I found many paper studying dynamic level generation from agent feedback and methods that scale to large game environments. I believe this is a scholarship problem with the UED community at large rather than a problem with specific authors. Level generation has been studied widely for decades, but many of the proposed methods have not been applied to UED settings where RL agent training is in the loop. However, I do think the paper should cite these related ideas and provide context to the novelty of the proposed work, which is only novel in a UED context. The authors should also consider citing the POET paper as it inspired this research area.

**Limitations:**

- The domain in the paper is not representative of all domains that could be used in the work. (For example, with respect to stochasticity or or continuous environments.) The current writing does not discuss domains where the methods presented could not be applied, nor gives an avenue for applying dynamic environment generation to more complex domains. I encourage the authors to expand the limitations section more. It's not clear to me how to apply the method and receive benefit on domains other than mini grid. Many domains seem like they would have no benefits (such as those with full observability or observability requiring knowledge of all environment free parameters).

**Paper Formatting Concerns:**

- The expectation on line 89 seems to have weird formatting, e.g., the M looks like it should be a subscript, and there is an unmatched right bracket ].
- The format appears to have been modified slightly on lines 72-73, as those lines appear to intersect with the page number.

**Quality:**

3

**Strengths And Weaknesses:**

- The paper provides a clear review of the UED problem in the context of reinforcement learning and curriculum learning’s history.
- I feel figure 1 is not as impactful as it could be. It would be better if the first figure gave an overview of the proposed method, or contrasted the approach with existing methods. This is clear and easy to do for dynamic environment generation (as done in Figure 2). For the MNA, a figure highlighting the intuition of the new regret approximation would improve the paper.
- The paper fails to cite the seminal paper in this space “Paired Open-Ended Trailblazer” from Wang et. al. Many of the ideas in the PAIRED paper formalized concepts of this work.
- The authors only test their proposed methods on the minigrid domain. To convince readers of the paper that the results generalize, more domains are needed.
- This paper feels like two separate papers stuck together. Instead of two thorough papers, we have two half-papers without thorough evaluation. Both dynamic environment generation and the new proposed regret method are complicated enough, and produce sufficiently independent benefits, that I think splitting these contributions into two papers would be better. I'd rather see two separate contributions that explore each avenue more thoroughly. I'd particularly be interested in how dynamic environment generation could be applied to environments with complex entanglement of the observation space and environment parameterization. Solving that issue seems essential for the method to be generally applicable.
- The authors mention the problem of small changes in levels resulting in large changes in difficulty. This is a well studied problem in the Procedural Content Generation community. I refer the authors to “The Unexpected Consequence of Incremental Design Changes” from Sturtevant et. al. as a starting point.
- The authors mention small changes in levels producing large changes in difficulty, and give a few examples in the paper, but they don’t evaluate their method on environments where this effect occurs strongly. If this is a core argument in the paper, there should be some evidence that the method improves performance under these conditions. I recommend taking the snakebird environment from Sturtevant et. al. (see above) or Sokoban as an evaluation as both have been shown to have these properties.
- There has been some work in scaling environment generation via neural cellular automata. This seems like a natural baseline to compare dynamic environment generation against as the paper has results on 66x66 minigrid mazes. See “Arbitrarily Scalable Environment Generators via Neural Cellular Automata” from NeurIPS 2023. A simple baseline would be to replace the evolutionary environment generator of ACCEL with NCAs optimized via CMA-ES with a regret objective.
- The presentation of experimental results could be more thorough. For example, it is unclear how many trials were used in the experiments. It is also not stated what the error bars in the plots represent. Finally, there is no table listing the quantitative results of all experiments.
- Overall, if a claim is made in the paper, experiments should be designed to support the claim. Domains should be catered to study the proposed benefits of the method. I feel this paper has a disconnect between claims made and the experiments run.

---

> ### Author Rebuttal · Authors · 2025-07-31
>
> Thank you for your review, we really appreciate the detailed and thorough response you have given, especially your insight into similar work in the Procedural Content Generation community that we had not previously considered.
>
> ## Combining MNA and DEGen
>
> Whilst we understand why you have suggested  presenting MNA and DEGen as two separate papers, we feel these two contributions are too closely linked for this. One of the core goals of this research was to enable denser reward signals for RL-training a level generator to reduce the difficulty of credit assignment. Previous work relying on a generator that fully generates the level initially only allowed for a single sparse reward at the final generator time step. By implementing DEGen, we can instead assign rewards throughout the level generator trajectory. In order to maximally leverage this advantage, a dense regret approximation was necessary. However in practice, we found that existing regret approximation metrics, such as PVL and MaxMC, performed poorly when used to train a DEGen adversary. This indicates that although these metrics outperform naive DR when used to curate levels generated through random level generation/mutation, when directly optimised for, these metrics perform poorly. This directly necessitated an improved regret approximation (MNA) that performs well in both with the curation-based methods, and with an RL-trained level-generator like in DEGen.
>
> ## Evaluation Environments
>
> The suggestions of Snakebird and Sokoban as environments that exhibit large changes in difficulty with small changes to the environment was useful. Based on this suggestion, we have begun to run experiments in Sokoban with some initial promising results. However, we do also believe that the provided key-based minigrid environment also strongly exhibits this property. Given two near-identical levels, but with one having a door blocking the path to the goal, and one without, a relatively small change of environment has induced a large change in difficulty. It is not simply that the optimal solution in the former will be longer, but to solve the former, the agent will have had to learn how to find the key, and how to use the key to unlock the door (i.e. learn a new class of solution).
>
> ## Experimental Details
>
> Each experiment was run over eight seeds, and the error bars shown are the 95% confidence intervals of these seeds. We can also include a detailed table of the quantitative results in the Appendix.
>
> # Questions
>
> ***Some details of dynamic environment generation confuse me …***
>
> DEGen is usable for methods with multiple agents. If both agents are running in parallel in the environment: after each protagonist/antagonist step, any ungenerated parts of the environment that are observed by either agent can be generated by the adversary. We have done some experimentation with the PAIRED objective with the DEGen adversary, so it is definitely possible to use methods that require multiple agents.
>
> ***In the derivation of MNA, the authors assume a deterministic environment …***
>
> In its current form, MNA is limited to deterministic environments given Eqn 6 does not necessarily hold in stochastic environments. However, as far as we are aware, existing UED work only focuses on deterministic environments. Stochastic environments greatly increase the difficulty of approximating the optimal return, as it becomes difficult to determine whether a given trajectory resulted in a high return because of strong actions or good luck with environmental randomness. E.g. a metric like MaxMC would likely break down in stochastic environments as it would score highly on levels that have a high probability of low reward and a low probability of high reward based purely on environment dynamics.
>
> ***The dynamic environment generation seems to be only applicable in domains where strong partial observability is present …***
>
> DEGen will only differ to methods that fully-generate the level up front in domains that are strongly partially observable. However, we feel there are sufficient domains where this holds for DEGen to have benefits. For example, robotics domains relying on local, limited range sensors. Also, many games will fall into this category, such as Crafter [1], or domains like side-scrolling platformers.
>
> ***In the paper, the minigrid levels are closely connected to observations …***
>
> It is a limitation of DEGen that in certain environments it may be more difficult to map environment parameterisations to agent observations. We will add a discussion of this to the limitations section of the paper.
>
>
> [1] Hafner, Danijar. "Benchmarking the spectrum of agent capabilities." arXiv preprint arXiv:2109.06780 (2021).

---

> ### Comment · Reviewer_7GXd · 2025-08-04
>
> Please do your best to post (even partial) results from the Sokoban experiments before the rebuttal period ends. I think adding a second domain would greatly strengthen the paper.
>
> If you could, please post the limitation discussion you propose to add to the paper regarding complex mappings between observation spaces and level representation. I believe solving this problem is crucial for the proposed method to be applied to any real world domain (robotics, web-browsing, etc). Eventually, UED methods should move away from the current focus on games and towards real world applications. This limitation is an obstacle in moving the proposed method in that direction.
>
> The authors should add some discussion to the limitations for the field on stochastic versus determinist environments. This also creates an interesting research opportunity a reader may gleam from your paper.
>
> I now see the authors point in how the two proposed methods are coupled. The writing of the paper needs to be strengthened to highlight this and motivate the paper better. Understanding why you need both components is key to understanding the motivations in the paper.
>
> I disagree with the authors that mini-grid is a good domain to observe large behavioral changes from small environment changes. While most domains exhibit this property in some capacity, the effects vary greatly on how frequently this occurs. Consider for example a random change of tile in a mini-grid level. For most of the level, a tile change would not change an agent's behavior. However, some key tiles greatly affect performance. This is something that could be quantified by analyzing tile changes in levels on different environments and seeing the effect on level completion. I expect that Sokoban would have a significantly larger effect on performance than mini-grid.
>
> I'd still like to see stronger baselines. There are many approaches to level generation that scale to large levels (NCA, WFC, etc). Given the interchangeable nature of UED methods, it should be possible to add another method designed to generate larger levels.

---

> ### Author Response · Authors · 2025-08-09
>
> # Sokoban Environment
>
> Below we have included the zero-shot solve rate on a selection of Sokoban levels. We have used the first 10 levels of Sokoban Jr 1 that had optimal solutions with less that 100 moves (it is unclear why, but the csv download we used for these levels did not include Sokoban_Jr_1_01). The table below shows the mean and standard error of three seeds for each UED method.
>
> | Level           | DR          | SFL         | PVL - PLR   | MaxMC - PLR   | MNA - PLR   | MNA - DEGen   |
> |:----------------|:------------|:------------|:------------|:--------------|:------------|:--------------|
> | Sokoban_Jr_1_02 | 0.00 ± 0.00 | 0.00 ± 0.00 | 0.43 ± 0.09 | 0.97 ± 0.03   | 0.87 ± 0.03 | **1.00 ± 0.00**   |
> | Sokoban_Jr_1_03 | 0.00 ± 0.00 | 0.00 ± 0.00 | 0.07 ± 0.07 | 0.80 ± 0.15   | 0.27 ± 0.07 | **0.97 ± 0.03**   |
> | Sokoban_Jr_1_04 | 0.00 ± 0.00 | 0.00 ± 0.00 | 0.00 ± 0.00 | 0.07 ± 0.07   | **0.70 ± 0.17** | 0.10 ± 0.10   |
> | Sokoban_Jr_1_05 | 0.00 ± 0.00 | 0.00 ± 0.00 | 0.07 ± 0.07 | **0.70 ± 0.30**   | 0.37 ± 0.27 | 0.47 ± 0.29   |
> | Sokoban_Jr_1_06 | 0.00 ± 0.00 | 0.00 ± 0.00 | 0.33 ± 0.23 | 0.83 ± 0.03   | 0.20 ± 0.12 | **1.00 ± 0.00**   |
> | Sokoban_Jr_1_07 | 0.00 ± 0.00 | 0.00 ± 0.00 | 0.13 ± 0.09 | 0.17 ± 0.17   | 0.30 ± 0.25 | **0.57 ± 0.30**   |
> | Sokoban_Jr_1_08 | 0.00 ± 0.00 | 0.00 ± 0.00 | 0.00 ± 0.00 | 0.00 ± 0.00   | 0.00 ± 0.00 | **0.70 ± 0.30**   |
> | Sokoban_Jr_1_09 | 0.00 ± 0.00 | 0.00 ± 0.00 | 0.17 ± 0.07 | 0.90 ± 0.06   | 0.67 ± 0.03 | **0.97 ± 0.03**   |
> | Sokoban_Jr_1_12 | 0.00 ± 0.00 | 0.00 ± 0.00 | 0.00 ± 0.00 | 0.00 ± 0.00   | 0.00 ± 0.00 | **0.30 ± 0.12**   |
> | Sokoban_Jr_1_16 | 0.00 ± 0.00 | 0.00 ± 0.00 | 0.13 ± 0.03 | **1.00 ± 0.00**   | 0.80 ± 0.12 | 0.97 ± 0.03   |
> | Mean            | 0.00 ± 0.00 | 0.00 ± 0.00 | 0.13 ± 0.03 | 0.54 ± 0.03   | 0.42 ± 0.05 | **0.70 ± 0.07**   |
>
> ## Environment Details
>
> As in standard Sokoban, the agent aims to get all boxes to their storage locations. The agent receives a sparse reward when completing a level proportional to how many steps were required to complete the level. As previously discussed, DEGen is limited to environments with partial observability - therefore we have implemented a minigrid-style agent for Sokoban, with a 5x5 observation space ahead of the agent, and an action space of {turn left, turn right, move forward}. The agent also has a {reset} action that resets the agent and boxes to their starting locations.
>
> We used 9x9 levels for training - for DR, PLR and SFL, the random level generator generated levels that had 15 walls, and between 1 and 10 boxes. For DEGen, the generator could fill each newly observed cell in the environment with an empty square/wall/box/storage location/box on storage location. For all methods, we used identical hyperparameters to the key-minigrid environment, except for training for 30k PPO updates, rather than 15k.
>
> ## Results
>
> From these preliminary results, it is clear the MNA-DEGen does consistently outperform the baselines. We have shown that even at a small level size (9x9), DEGen outperforms methods relying on random level generation. It is clear that as the level size and number of boxes increases, it is likely infeasible to use methods that rely on random generation, as with a greater number of boxes, the probability of at least one box being impossible to move to a storage location increases. Therefore it is likely necessary to use methods relying on trained generators, such as DEGen.
>
> **Limitations of Current Results**
>
> Due to the limited time of this discussion period, we have not been fully able to assess the Sokoban environment. We have not been able to do any hyperparameter tuning, and have only been able to assess 3 seeds for only a subset of the baselines used in both minigrid and key-minigrid. The results in their current state do highlight the merits of MNA-DEGen, but we will be able to perform thorough testing for the final version on the paper.

---

> ### Author Response · Authors · 2025-08-09
>
> # Mapping between Observation Space and Level Representation - Limitations
>
> Whilst the results we have presented in this paper do show strong performance from DEGen compared to existing baselines that do not leverage the agent's partial observability, the domains presented in this paper present relatively simple mappings between the level representation and the agent's current observation. In more complex domains, such as Doom, or real-world robotics applications, it will be more complex to determine how specific environment parameters affect what the agent is currently observing. In order for UED to bridge the gap from the current set of game domains to real-world applications, it would be necessary for DEGen or DEGen-like methods to address this limitation. We believe that World Models [1] represent a promising direction for future research to address this. In its current form, DEGen relies on both the agent and the generator interacting with a fixed environment, where the environment has an explicit mapping between the agent's observation and the level parameters, and the generator is only able to generate the level where the agent has observed. However, a world model guided by a regret approximation such as MNA would represent a generator that could directly generate observations for the agent. Rather than relying on explicit mapping between level and observation, this mapping could be learnt with environment data when training the world model. Whilst the training of a world model would add additional computational cost to the training process, it would enable DEGen methodology to be applied to substantially more complex environments. With the advent of highly general world models such as the Genie series of world models [2], this could represent a path to training highly general policies that are effective in a wide variety of applications.
>
> # Stochastic Environments - Limitations
>
> Current UED work has focused primarily on deterministic environments as current methods of approximating regret will likely break down in highly stochastic environments. Approximating regret in stochastic environments represents a significant challenge compared to deterministic environments. Many current UED methods rely on the maximum return of an agent on a given level to approximate the optimal return [3, 4] when computing regret, but in stochastic environments, this may overestimate the expected optimal return if this maximum was achieved when the agent got "lucky" with environmental stochasticity. Similarly, methods relying on value-loss regret approximations may break down, as value-loss could also result from environmental stochasticity, rather than just from value function error or policy action selection. Future research may need to focus on presenting regret approximations that are suitable for highly stochastic environments.
>
> # Baselines
>
> We do agree that current UED baselines are weak for larger level sizes, and we appreciate your suggestions for alternatives. We will aim to present further baselines in the final version of the paper. However, we do also feel that these methods may be deserving of specific research, given as far as we know, these methods have not been previously applied in a UED context.
>
> [1] Ha, David, and Jürgen Schmidhuber. "World models." arXiv preprint arXiv:1803.10122 2.3 (2018).
>
> [2] Bruce, Jake, et al. "Genie: Generative interactive environments." Forty-first International Conference on Machine Learning. 2024.
>
> [3] Dennis, Michael, et al. "Emergent complexity and zero-shot transfer via unsupervised environment design." Advances in neural information processing systems 33 (2020): 13049-13061.
>
> [4] Jiang, Minqi, et al. "Replay-guided adversarial environment design." Advances in Neural Information Processing Systems 34 (2021): 1884-1897.

---

### Decision · Program_Chairs · 2025-09-17

**Decision:**

Accept (poster)

**Comment:**

The paper had two accepts, one borderline accept and one strong reject (at the decision deadline). The authors addressed many of the points in the strong reject review, yet the reviewer has not replied to the authors or justified their score on time. I therefore chose to ignore this review predicting that the reviewer will most likely increase their score (as they did post the decision deadline). Given that the majority of reviews recommended acceptance, and found the paper to be well motivated with creative ideas and good experiments, I recommend accepting it.